# Epistemic Neural Networks

**Ian Osband**,[*] **Zheng Wen, Seyed Mohammad Asghari,**
**Vikranth Dwaracherla, Morteza Ibrahimi, Xiuyuan Lu, and Benjamin Van Roy**
Google DeepMind, Efficient Agent Team, Mountain View
`{ian.osband, m.ibrahimi}@gmail.com`
`{zhengwen,smasghari,vikranthd,lxlu,benvanroy}@google.com`

## Abstract

Intelligent agents need to know what they don't know, and this capability can be evaluated through the quality of *joint* predictions. In principle, ensemble methods can produce effective joint predictions, but the compute costs are prohibitive for large models. We introduce the *epinet*: an architecture that can supplement any conventional neural network, including large pretrained models, and can be trained with modest incremental computation to estimate uncertainty. With an epinet, conventional neural networks outperform large ensembles of hundreds or more particles, and use orders of magnitude less computation. The epinet does not fit the traditional framework of Bayesian neural networks, so we introduce the *epistemic neural network* (ENN) as a general interface for models that generate joint predictions.

## 1 Introduction

Consider a conventional neural network trained to predict whether a random person would classify a drawing as a 'rabbit' or a 'duck'. As illustrated in Figure 1, given a single drawing, the network outputs a *marginal* prediction that assigns probabilities to the two classes. If the probabilities are each 0.5, it remains unclear whether this is because labels sampled from random people are equally likely, or whether the neural network would learn a single class if trained on more data. Conventional neural networks do not distinguish these cases, even though it can be critical for decision making systems to know what they do not know. This capability can be assessed through the quality of *joint* predictions (Wen et al., 2022).

The two tables to the right of Figure 1 represent possible joint predictions that are each consistent with the network's uniform marginal prediction. These joint predictions are over *pairs* of labels for the same image, $(y_1, y_2) \in \{R, D\} \times \{R, D\}$. For any such joint prediction, Bayes' rule defines a conditional prediction for $y_2$ given $y_1$. The first table indicates inevitable uncertainty that would not be resolved through training on additional data; conditioning on the first label does not alter the prediction for the second. The second table indicates that additional training should resolve uncertainty; conditioned on the first label, the prediction for the second label assigns all probability to the same outcome as the first.

Figure 1 presents the toy problem of predictions across two identical images as a simple illustration of these types of uncertainty. The observation that joint distributions express whether uncertainty is resolvable extends more generally to practical cases, where the inputs differ, or where there are more than two simultaneous predictions (Osband et al., 2022a).

Bayesian neural networks (BNNs) offer a statistically-principled way to make effective joint predictions, by maintaining an approximate posterior over the weights of a base neural network. Asymptotically these can recover the exact posterior, but the computational costs

---

[*]Contact `ian.osband@gmail.com`

37th Conference on Neural Information Processing Systems (NeurIPS 2023).

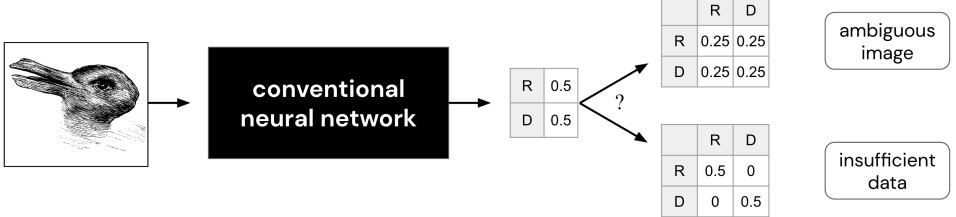

Figure 1: Conventional neural nets generate marginal predictions, which do not distinguish genuine ambiguity from insufficiency of data. Joint predictions can make this distinction.

are prohibitive for large models (Welling and Teh, 2011). Ensemble-based BNNs offer a more practical approach by approximating the posterior distribution with an ensemble of statistically plausible networks that we call *particles* (Osband and Van Roy, 2015; Lakshminarayanan et al., 2017). While the quality of joint predictions improves with more particles, practical implementations are often limited to at most tens of particles due to computational constraints.

In this paper, **we introduce an approach that outperforms ensembles of hundreds of particles at a computational cost less than that of two particles**. Our key innovation is the *epinet*: a network architecture that can be added to any conventional neural network to estimate uncertainty. Figure 2 offers a preview of results presented in Section 6, where we compare these approaches on ImageNet. The quality of the ResNet's marginal predictions – measured by classification error or marginal log-loss – does not change much if supplemented with an epinet. However the epinet-enhanced ResNet dramatically improves the quality of *joint* predictions, as measured by the joint log-loss, outperforming the ensemble of 100 particles, with total parameters less than 2 particles. Prior work has shown the importance of *joint* predictions in driving effective decisions for a broad class of problems, including combinatorial decision problems and sequential decision problems (Wen et al., 2022; Osband et al., 2022a).

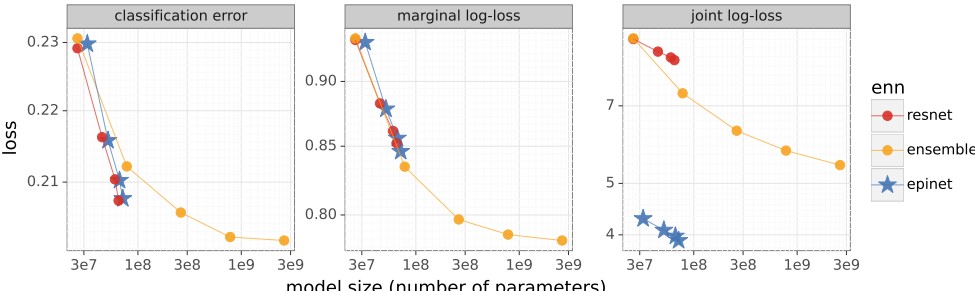

Figure 2: Quality of marginal and joint predictions across models on ImageNet (Section 6).

The epinet does not fit into the traditional framework of BNNs. In particular, it does not represent a distribution over base neural network parameters. To accommodate development of the epinet and other approaches that do not fit the BNN framework, we introduce the concept of *epistemic neural networks* (ENNs). We establish that all BNNs are ENNs, but there are useful ENNs such as the epinet, that are not BNNs.

## 2 Related work

Our research builds on the literature in Bayesian deep learning (Hinton and Van Camp, 1993; Neal, 2012). BNNs represents epistemic uncertainty via approximating the posterior distribution over parameters of a base neural network (Der Kiureghian and Ditlevsen, 2009; Kendall and Gal, 2017). A challenge is the computational cost of posterior inference, which becomes intractable even for small networks (MacKay, 1992), and even approximate SGMCMC becomes prohibitive for large scale models (Welling and Teh, 2011).

Tractable methods for approximate inference has renewed interest in BNNs. Variational approaches such as Bayes by backprop (Blundell et al., 2015) use an evidence-based lower bound (ELBO) to approximate the posterior distribution, and related approaches use the same objective with more expressive weight distributions (Louizos and Welling, 2017). One

influential line of work claims that MC dropout can be viewed as one such approach (Gal and Ghahramani, 2016), although subsequent papers have noted that the quality of this approximation can be very poor (Osband, 2016; Hron et al., 2017). As of 2022, perhaps the most popular approach is ensemble-based, with an ensemble of models, each referred to as a *particle*, trained in parallel so that they together approximate a posterior distribution (Osband and Van Roy, 2015; Lakshminarayanan et al., 2017).

Ensemble-based BNNs train multiple particles independently. This incurs computational cost that scales with the number of particles. A thriving literature has emerged that seeks the benefits of large ensembles at lower computational cost. Some approaches only ensemble parts of the network, rather than the whole (Osband et al., 2019; Havasi et al., 2020). Others introduce new architectures to directly incorporate uncertainty estimates, often inspired by connections to Gaussian processes (Malinin and Gales, 2018; Charpentier et al., 2020; Liu et al., 2020; van Amersfoort et al., 2021). Others perform Bayesian inference more directly in the function space in order to sidestep issues relating to overparameterization (Sun et al., 2019).

In general, research in Bayesian deep learning has focused more on developing methodology than unified evaluation (Osband et al., 2022a). Perhaps because the potential benefits of BNNs are so far-reaching, different papers have emphasized improvements in classification accuracy (Wilson, 2020), expected calibration error (Ovadia et al., 2019), OOD performance (Hendrycks and Dietterich, 2019), active learning (Gal et al., 2017) and decision making (Osband et al., 2019). However, in each of these settings it generally is possible to obtain improvements via methods that do not aim to approximate posterior distributions. Perhaps for this reason, there has been a recent effort to refocus evaluation on how well methods actually approximate 'gold standard' Bayes posteriors (Izmailov et al., 2021).

Our work on ENNs is motivated by the importance of joint predictions in driving decision, exploration, and adaptation (Wang et al., 2021; Wen et al., 2022; Osband et al., 2022a). This line of research, which we build on in Section 3.1, establishes a sense in which joint predictions are both necessary and sufficient to drive decisions. Effectiveness of ENN designs can be assessed through the quality of joint predictions. This perspective allows us to consider approaches beyond those accommodated by the BNN framework. As we will demonstrate, this can lead to significant improvements in performance.

## 3 Epistemic neural networks

A conventional neural network is specified by a parameterized function class $f$, which produces a vector-valued output $f_\theta(x)$ given parameters $\theta$ and an input $x$. The output $f_\theta(x)$ assigns a corresponding probability $\hat{P}(y) = \exp\left((f_\theta(x))_y\right) / \sum_{y'} \exp\left((f_\theta(x))_{y'}\right)$ to each class $y$. For shorthand, we write such class probabilities as $\hat{P}(y) = \text{softmax}(f_\theta(x))_y$. We refer to a predictive class distribution $\hat{P}$ produced in this way as a *marginal prediction*, as it pertains to a single input $x$.

An ENN architecture, on the other hand, is specified by a pair: a parameterized function class $f$ and a reference distribution $P_Z$. The vector-valued output $f_\theta(x, z)$ of an ENN depends additionally on an *epistemic index* $z$, which takes values in the support of $P_Z$. Typical choices of the reference distribution $P_Z$ include a uniform distribution over a finite set or a standard Gaussian over a vector space. The index $z$ is used to express epistemic uncertainty. In particular, variation of the network output with $z$ indicates uncertainty that might be resolved by future data. As we will see, the introduction of an epistemic index allows us to represent the kind of uncertainty required to generate useful joint predictions.

Given inputs $x_1,...,x_\tau$, a joint prediction assigns a probability $\hat{P}_{1:\tau}(y_{1:\tau})$ to each class combination $y_1,...,y_\tau$. While conventional neural networks are not designed to provide joint predictions, joint predictions can be produced by multiplying marginal predictions:

$$\hat{P}_{1:\tau}^{\text{NN}}(y_{1:\tau}) = \prod_{t=1}^{\tau} \text{softmax}(f_\theta(x_t))_{y_t}. \tag{1}$$

However, this representation models each outcome $y_{1:\tau}$ as independent and so fails to distinguish ambiguity from insufficiency of data. ENNs address this by enabling more

expressive joint predictions through integrating over epistemic indices:

$$\hat{P}_{1:\tau}^{\mathrm{ENN}}(y_{1:\tau}) = \int_z P_Z(dz) \prod_{t=1}^{\tau} \mathrm{softmax}\left(f_\theta(x_t, z)\right)_{y_t}.$$ (2)

This integration introduces dependencies so that joint predictions are not necessarily just the product of marginals. Figure 3 provides a simple example of how two different ENNs can use the epistemic index to distinguish the sorts of uncertainty described in Figure 1.

In Figure 3(a) the ENN makes marginal predictions that do not vary with $z$, and so the resultant joint predictions are simply the independent product of marginals. This corresponds to an 'aleatoric' or 'irreducible' form of uncertainty that cannot be resolved with data. On the other hand, Figure 3(b) shows an ENN that makes predictions depending on the sign of the epistemic index. This corresponds to 'epistemic' or 'reducible' uncertainty that can be resolved with data. In this case, integrating the 2x2 matrix over $z$ produces a diagonal matrix with 0.5 in each diagonal entry. As such, Figure 3 shows how an ENN can use the epistemic index to distinguish the two joint distributions of Figure 1.

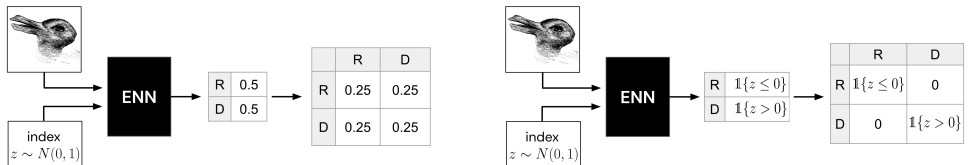

   (a) An ENN indicating an ambiguous image.    (b) An ENN indicating insufficient data.

Figure 3: An ENN can incorporate the epistemic index $z \sim P_Z$ into its joint predictions. This allows an ENN to differentiate inevitable ambiguity from data insufficiency.

## 3.1 Evaluating ENN performance

Marginal log loss (also known as cross-entropy loss) is perhaps the most widely used evaluation metric in machine learning. For a single input $x$, if a neural network generates a prediction $\hat{P}$ and the label turns out to be $y$, then the sample log loss is the form $-\ln \hat{P}(y)$. We say that this is a *marginal* loss because it only looks at the quality of a prediction over a single (input, output) pair. As we will discuss, minimizing marginal log loss does not generally lead to performant downstream decisions. Good decisions often require good *joint* predictions.

To formulate a generic decision problem, consider a reward function $r$ that maps an action $a \in \mathcal{A}$ and $\tau$ labels $y_{1:\tau}$ to a reward $r(a, y_{1:\tau}) \in [0, 1]$. Given an exact posterior predictive $P_{1:\tau}$, consider as an objective maximization of the expected reward $\sum_{y_{1:\tau}} P_{1:\tau}(y_{1:\tau})r(a, y_{1:\tau})$. The optimal decision can be approximated based on an ENN's prediction $\hat{P}_{1:\tau}$ by choosing an action $a$ that maximizes $\sum_{y_{1:\tau}} \hat{P}_{1:\tau}(y_{1:\tau})r(a, y_{1:\tau})$. The following theorem is formalized and proved in Appendix B.

**Theorem 1.** [informal] *There exists a decision problem and an ENN that attains small expected marginal log loss such that actions generated using the ENN perform no better than random guessing.*

Theorem 1 indicates that minimizing marginal log loss does not suffice to support effective decisions. The key to this insight is that marginal predictions do not distinguish ambiguity from insufficiency of data. However, this can can be addressed by instead considering the *joint* log loss. Given $\tau$ data pairs and a joint prediction $\hat{P}_{1:\tau}$, we can consider the joint log loss $-\ln \hat{P}_{1:\tau}(y_{1:\tau})$ in exactly the same way that we looked at the marginal log loss. We formalize our next result in Appendix B.

**Theorem 2.** [informal] *For any decision problem, any ENN that attains small expected joint log loss leads to actions that attain near optimal expected reward.*

Theorems 1 and 2 highlight the importance of joint predictions in driving decisions. Since we want machine learning systems to drive effective decisions, we will assess the performance

of ENNs by comparing their joint log loss.[2] It is important to note that we will consider this joint loss as a method for assessing quality of a *trained* ENN. We have not yet discussed how particular forms of ENNs are trained, which will generally be up to the algorithm designer. Section 4 provides further detail on the specific architecture and training loss for the epinet ENN we develop in this paper.

## 3.2 ENNs versus BNNs

A base neural network $f$ defines a class of functions. Each element $f_\theta$ of this class is identified by a vector $\theta$ of parameters, which specify weights and biases. An ENN or BNN is designed with respect to a specific base network, and seek to express uncertainty while learning a function in this class. We will formally define what it means for an ENN or BNN to be defined with respect to a base network. We will then establish results which indicate that, with respect to any base network, all BNNs can be expressed as ENNs but not vice versa.

Consider a base neural network $f$ which, given parameters $\theta$ and input $x$, produces an output $f_\theta(x)$. A typical BNN is specified by a pair: a base network $f$ and a parameterized sampling distribution $p$. Given parameters $\nu$, a sample $\hat{\theta}$ can be drawn from the distribution $p_\nu$ to generate a function $f_{\hat{\theta}}$. Approaches such as stochastic gradient MCMC, deep ensembles, and dropout can all be framed in this way. For example, with a deep ensemble, $\hat{\theta}$ comprises parameters of an ensemble particle and $p_\nu$ is the distribution represented by the ensemble, which assigns probability to a finite set of vectors, each associated with one ensemble particle. For any inputs $x_1, \ldots, x_\tau$, by sampling many functions $(f_{\hat{\theta}^k} : k = 1, \ldots, K)$, a BNN can be used to approximate the corresponding joint distribution over labels $y_1, \ldots, y_\tau$, according to $\hat{P}(y_{1:\tau}) = \frac{1}{K} \sum_{k=1}^{K} \mathbf{1}((f_{\hat{\theta}^k}(x_1), \ldots, f_{\hat{\theta}^k}(x_\tau)) = y_{1:\tau})$.

We say the BNN $(f, p)$ is *defined with respect to* its base network $f$. We say an ENN $(f', P_Z)$ is *defined with respect* to a base network $f$ if, for any base network parameters $\theta$, there exist ENN parameters $\theta'$ such that $f'_{\theta'}(\cdot, z) = f_\theta$ almost surely with respect to $P_Z$. Intuitively, being defined with respect to a particular base network means that the BNN or ENN is designed to learn any function within the class characterized by the base network.

We say that a BNN $(f, p)$ is *expressed as* an ENN $(f', P_Z)$ if, for all $\nu$, $\tau$, and inputs $x_{1:\tau}$, there exists $\theta'$ such that for $\hat{\theta} \sim p_\nu, z \sim P_z$,

$$(f_{\hat{\theta}}(x_1), \ldots, f_{\hat{\theta}}(x_\tau)) \stackrel{d}{=} (f'_{\theta'}(x_1, z), \ldots, f'_{\theta'}(x_\tau, z)). \tag{3}$$

This condition means that the ENN and BNN make the same joint predictive distributions at all inputs.

We say that an ENN $(f', P_Z)$ is *expressed as* a BNN if, for all $\theta'$, $\tau$, and inputs $x_1, \ldots, x_\tau$, there exists a posterior distribution $\nu$ such that (3) holds. Intuitively, one architecture is expressed as the other if the latter can represent the same distributions over functions. The following result, established in Appendix C, asserts that any BNN can be expressed as an ENN but not every ENN can be expressed as a BNN.

**Theorem 3.** *For all base networks $f$, any BNN defined with respect to $f$ can be expressed as an ENN defined with respect to $f$. However, there exists a base network $f$ and ENN defined with respect to $f$ that can not be expressed as a BNN defined with respect to $f$.*

In supervised learning, de Finetti's Theorem implies that if a sequence of data pairs is exchangeable then they are i.i.d. conditioned on a latent random object (de Finetti, 1929). BNNs use base network parameters $\theta$ as the object, while ENNs focus on the function $g_*$ itself, without concerning the underlying parameters. ENNs serve as computational mechanisms to approximate the posterior distribution of $g_*$, allowing functions beyond the base network class to represent uncertainty and allowing better trade-offs between computation and prediction quality. The epinet is an example of an ENN that cannot be expressed as a BNN with the same base network, and showcases the benefits of the ENN interface beyond BNNs.

---

[2]In problems with high-dimensional inputs, the number of inputs $x_1, .., x_\tau$ sampled uniformly from the input distribution, may have to be very large to distinguish ENNs in ways marginal log loss does not (Osband et al., 2022b). To sidestep prohibitive computational costs we use dyadic sampling in our empirical evaluation and review these details in Appendix F.

# 4  The epinet

This section introduces the *epinet*, which can supplement any conventional neural network to make a new kind of ENN architecture. Our approach reuses standard deep learning components and training algorithms, as outlined in Algorithm 1. As we will see, it is straightforward to add an epinet to any existing model, even one that has been pretrained. The key to successful application of the epinet comes in the design of the network architecture and loss functions.

---

**Algorithm 1** ENN training via SGD

**Inputs:**

| | |
|---|---|
| dataset | training examples $\mathcal{D} = \{(x_i, y_i, i)\}_{i=1}^{N}$ |
| ENN | network $f$, reference $P_Z$, initialization $\theta_0$ |
| loss | $\ell$ evaluates example $(x_i, y_i, i)$ for index $z$ |
| batch size | data samples $n_B$, index samples $n_Z$ |
| optimizer | update rule and number of iterations $T$ |

**Returns:**

| | |
|---|---|
| $\theta_T$ | parameter estimates for the ENN. |

1: **for** $t$ in $0, ..., T-1$ **do**
2:     sample data $\tilde{\mathcal{I}} = i_1, .., i_{n_B} \sim \text{Unif}(\{1, .., N\})$.
3:     sample indices $\tilde{\mathcal{Z}} = z_1, .., z_{n_Z} \sim P_Z$.
4:     compute $\texttt{grad} \leftarrow \nabla_{\theta|\theta=\theta_t} \sum_{z \in \tilde{\mathcal{Z}}} \sum_{i \in \tilde{\mathcal{I}}} \ell(\theta, z, x_i, y_i, i)$.
5:     update $\theta_{t+1} \leftarrow \texttt{optimizer}(\theta_t, \texttt{grad})$

---

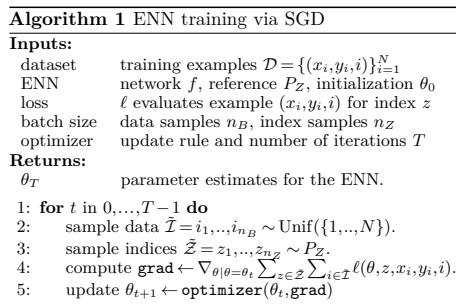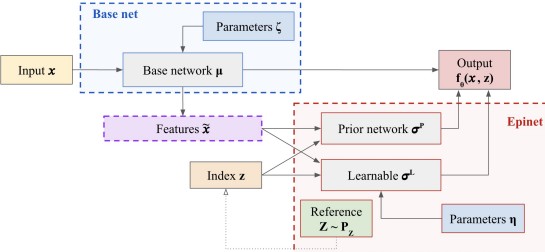

Figure 4: Epinet network architecture.

## 4.1  Architecture

Consider a conventional neural network as the *base network*. Given base parameters $\zeta$ and an input $x$, the output is $\mu_\zeta(x)$. For a classification model, the class probabilities would be $\text{softmax}(\mu_\zeta(x))$. An epinet is a neural network with privileged access to inputs and outputs of activation units in the base network. A subset of these inputs and outputs, which we call *features* $\phi_\zeta(x)$, are taken as input to the epinet along with an epistemic index $z$. For example, these features might be the last hidden layer in a ResNet. The epistemic index is sampled from a standard Gaussian distribution in dimension $D_Z$. For epinet parameters $\eta$, the epinet outputs $\sigma_\eta(\phi_\zeta(x), z)$. To produce an ENN, the output of the epinet is added to that of the base network, though with a "stop gradient":[3]

$$\underbrace{f_\theta(x, z)}_{\text{ENN}} = \underbrace{\mu_\zeta(x)}_{\text{base net}} + \underbrace{\sigma_\eta(\text{sg}[\phi_\zeta(x)], z)}_{\text{epinet}}. \tag{4}$$

We find that with this stop gradient, training dynamics more reliably produce models that perform well out of sample. The ENN parameters $\theta = (\zeta, \eta)$ include those of the base network $\zeta$ and epinet $\eta$. Due to the additive structure of the epinet, multiple samples of the ENN can be obtained with only one forward pass of the base network. Where the epinet is much smaller than the base network this can lead to significant computational savings.

Before training, variation of the ENN output $f_\theta(x, z)$ as a function of $z$ reflects prior uncertainty in predictions. Since the base network does not depend on $z$, this variation must derive from the epinet. In our experiments, we induce this initial variation using *prior networks* (Osband et al., 2018). In particular, for $\tilde{x} := \text{sg}[\phi_\zeta(x)]$, our epinets take the form

$$\underbrace{\sigma_\eta(\tilde{x}, z)}_{\text{epinet}} = \underbrace{\sigma_\eta^L(\tilde{x}, z)}_{\text{learnable}} + \underbrace{\sigma^P(\tilde{x}, z)}_{\text{prior net}}. \tag{5}$$

The prior network $\sigma^P$ represents prior uncertainty and has no trainable parameters. The learnable network $\sigma_\eta^L$ is typically initialized to output values close to zero, but is then trained so that the resultant sum $\sigma_\eta$ produces statistically plausible predictions for all probable values of $z$. Variations of a prediction $\sigma_\eta = \sigma_\eta^L + \sigma^P$ at an input $x$ as a function of $z$ indicate predictive epistemic uncertainty, just like the example from Figure 3(b).

Epinet architectures can be designed to encode inductive biases that are appropriate for the application at hand. In this paper we focus on a particularly simple form of architecture

---

[3]The "stop gradient" notation $\text{sg}[\cdot]$ indicates the argument is treated as fixed when computing a gradient. For example, $\nabla_\theta f_\theta(x, z) = [\nabla_\zeta \mu_\zeta(x), \nabla_\eta \sigma_\eta(\phi_\zeta(x), z)]$.

for $\sigma_\eta^L$ based around standard multi-layered perceptron (MLP) with Glorot initialization (Glorot and Bengio, 2010):

$$\sigma_\eta^L(\tilde{x}, z) := \mathtt{mlp}_\eta([\tilde{x}, z])^\top z \in \mathbb{R}^C, \tag{6}$$

where $\mathtt{mlp}_\eta$ is an MLP with outputs in $\mathbb{R}^{D_z \times C}$ and $[\tilde{x}, z]$ is a flattened concatenation of $\tilde{x}$ and $z$. Depending on the choice of hidden units, $\sigma_\eta^L$ can represent highly nonlinear functions in $\tilde{x}, z$ and thus allow for expressive joint predictions in high level features. The design, initialization and scaling the prior network $\sigma^P$ allows an algorithm designer to encode prior beliefs, and is essential for good performance in learning tasks. Typical choices might include $\sigma^P$ sampled from the same architecture as $\sigma^L$ but with different parameters.

## 4.2 Training loss function

While the epinet's novelty primarily lies in its architecture, the choice of loss function for training can also play a role in its performance. In many classification problems, the standard regularized log loss suffices:

$$\ell_\lambda^{\mathrm{XENT}}(\theta, z, x_i, y_i, i) := -\ln\left(\mathrm{softmax}(f_\theta(x_i, z))_{y_i}\right) + \lambda\|\theta\|_2^2. \tag{7}$$

Here, $\lambda$ is a regularization penalty hyperparameter, while other notation are as defined in Sections 3 and 4.1. This is the loss function we use for the experiments with the Neural Testbed (Section 5) and ImageNet (Section 6).

Image classification benchmarks often exhibit a very high signal-to-noise ratio (SNR); identical images are almost always assigned the same label. As demonstrated by Dwaracherla et al. (2022), when the SNR is not so high and varies significantly across inputs, it can be beneficial to randomly perturb the loss function via versions of the statistical bootstrap (Efron and Tibshirani, 1994). For example, a Bernoulli bootstrap omits each data pair with probability $p \in [0, 1]$, giving rise to a perturbed loss function:

$$\ell_{p,\lambda}^{\mathrm{XENT}}(\theta, z, x_i, y_i, i) := \begin{cases} \ell_\lambda^{\mathrm{XENT}}(\theta, z, x_i, y_i, i) & \text{if } c_i^T z > \Phi^{-1}(p) \\ 0 & \text{otherwise}, \end{cases} \tag{8}$$

where each $c_i$ is an independent random vector sampled uniformly from the unit sphere, and $\Phi(\cdot)$ is the cumulative distribution function of the standard normal distribution $N(0, 1)$.

Note that the loss functions defined in equation 7 and 8 only explicitly state the loss for a single input-label pair $(x_i, y_i)$ and a single epistemic index $z$. To compute a stochastic gradient, one needs to sample a batch of input-label pairs and a batch of epistemic indices, average the losses defined above, and then compute the gradient.

## 4.3 How can this work?

The epinet is designed to produce effective *joint* predictions. As such, one might expect the training loss to explicitly reflect this and be surprised that we use standard marginal loss functions such as $\ell_\lambda^{\mathrm{XENT}}$. Recall that a prediction $f_\theta(x, z)$ produced by an epinet is given by a trainable component $\mu_\zeta(x) + \sigma_\eta^L(\tilde{x}, z)$ perturbed by the prior function $\sigma^P(\tilde{x}, z)$, which has no trainable parameters. Minimizing $\ell_\lambda^{\mathrm{XENT}}$ can therefore be viewed as optimizing the *learnable* component $\mu_\zeta(x) + \sigma_\eta^L(\tilde{x}, z)$ with a perturbed loss function. Previous work has established that learning with prior functions can induce effective joint predictions (Osband et al., 2018; He et al., 2020; Dwaracherla et al., 2020, 2022), we extend this to the epinet.

To show this marginal loss function can lead to effective joint predictions, Theorem 4 proves that this epinet training procedure can mimic exact Bayesian linear regression. Although this paper focuses on classification, in this subsection we consider a regression problem because its analytical tractability facilitates understanding. To establish this result, we introduce a regularized squared loss perturbed by Gaussian bootstrapping:

$$\ell_{\sigma,\lambda}^{\mathrm{LSG}}(\theta, z, x_i, y_i, i) := (f_{\zeta,\eta}(x_i, z) - y - \sigma c_i^T z)^2 + \lambda\left(\|\zeta\|_2^2 + \|\eta\|_2^2\right). \tag{9}$$

Here, each $c_i$ is a context vector sampled uniformly from the unit sphere in $D_Z$ dimensions, in the same manner as $\ell_{p,\lambda}^{\mathrm{XENT}}$ (8).

We say that a dataset $\mathcal{D}$ is *generated by a linear-Gaussian model* if $g_{\nu_*}(x) = \nu_*^\top x$, $\nu_* \sim N(0, \sigma_0^2 I)$, and each data pair $(x_i, y_i) \in \mathcal{D}$ satisfies $y_i = g_{\nu_*}(x_i) + \epsilon_i$ where $\epsilon_{1:N}$ are i.i.d. according to $N(0, \sigma^2)$. We say an ENN is a *linear-Gaussian epinet* with parameters $\theta = (\zeta, \eta)$ if its trainable component is comprised of a linearly parameterized functions $\mu_\zeta(x) = \zeta^T x$ and $\sigma_\eta^L(\tilde{x}, z) = z^T \eta \tilde{x}$, with epinet input $\tilde{x} = x$, the prior function takes the form $\sigma^P(x, z) = \sigma_0 z^T P_0 x$, where each column of the matrix $P_0$ is independently sampled uniformly from the unit sphere, and the reference distribution is taken to be $P_z \sim N(0, I_{D_Z})$.

**Theorem 4.** *Let data $\mathcal{D} = \{(x_i, y_i, i)\}_{i=1}^N$ be generated by a linear-Gaussian model and $f$ be a linear-Gaussian epinet. Let $\hat{\theta} \in \arg\min_\theta \sum_{i=1}^N \int_z P_Z(dz) \ell_{\sigma,\lambda}^{\mathrm{LSG}}(\theta, z, x_i, y_i, i)$ with parameter $\lambda = \sigma^2 / (N\sigma_0^2)$. Then, conditioned on $(\mathcal{D}, c_{1:N}, P_0)$, $f_{\hat{\theta}}(\cdot, z)$ converges in distribution to $g_{\nu_*}$ as $D_Z$ grows, almost surely.*

This result, proved in Appendix D, serves as a 'sanity check' that an epinet trained with a standard loss function can approach optimal joint predictions as the epistemic index dimension grows. Although our analysis is limited to linear-Gaussian models, the loss function applies more broadly. Indeed, we next demonstrate that epinets and standard loss functions scale effectively to large and complex models.

Table 1: Summary of benchmark agents, full details in Appendix G.

| agent | description | hyperparameters |
|---|---|---|
| mlp | vanilla MLP | $L_2$ decay |
| ensemble | deep ensembles (Lakshminarayanan et al., 2017) | $L_2$ decay, ensemble size |
| dropout | Dropout (Gal and Ghahramani, 2016) | $L_2$ decay, network, dropout rate |
| bbb | Bayes by backprop (Blundell et al., 2015) | prior mixture, network, early stopping |
| hypermodel | hypermodel (Dwaracherla et al., 2020) | $L_2$ decay, prior, bootstrap, index dimension |
| ensemble+ | ensemble + prior functions (Osband et al., 2018) | $L_2$ decay, ensemble size, prior scale, bootstrap |
| sgmcmc | stochastic gradient MCMC (Welling and Teh, 2011) | learning rate, prior, momentum |
| epinet | MLP + MLP epinet (this paper) | $L_2$ decay, network, prior, index dimension |

## 5 The neural testbed

*The Neural Testbed* is an open-source benchmark that evaluates the quality of joint predictions in classification problems using synthetic data produced by neural-network-based generative models (Osband et al., 2022a). We use this as a unit test to sanity-check learning algorithms in a controlled environment and compare the epinet against benchmark approaches.

Table 1 lists the agents that we study as well as hyperparameters that we tune via grid search. For our `epinet` agent, we use base network $\mu_\zeta(x)$ that matches the baseline `mlp` agent. We take the features $\phi(x)$ to be a concatenation of the input $x$ and the last hidden layer of the base network. We initialize the learnable epinet $\sigma^L$ according to (6) with 2 hidden layers of 15 hidden units. The prior network $\sigma^P$ is initialized as an ensemble of $D_Z = 8$ networks each with 2 hidden layers of 5 hidden units in each layer, and combine the output by dot-product with index $z$. We push the details, together with open source code, to Appendix G.

Figure 5 examines the trade-offs between statistical loss and computational cost for epinet against benchmark agents. The bars indicate standard error, estimated over the testbed's random seeds. After tuning, all agents perform similarly in marginal prediction, but the epinet is able to provide better joint predictions at lower computational cost. We first compare the performance of the epinet (blue) against that of an ensemble (red) as we grow the number of particles. **We show the epinet is able to perform much better than an ensemble with 100 particles, with a model less than twice the size of a single particle.** These results are still compelling when we compare against `ensemble+`, which includes prior functions, and which is necessary for good performance in low data regimes included in testbed evaluation. Included in this plot are the other agents `bbb`, `dropout` and `hypermodel` agents. The dashed line indicates performance of `sgmcmc` agent, which can asymptotically obtain the Bayes optimal solution, but at a much higher computational cost.

## 6 ImageNet

Benefits of the `epinet` scale up to more complex datasets and, in fact, become more substantial. This section focuses on experiments involving the ImageNet dataset (Deng

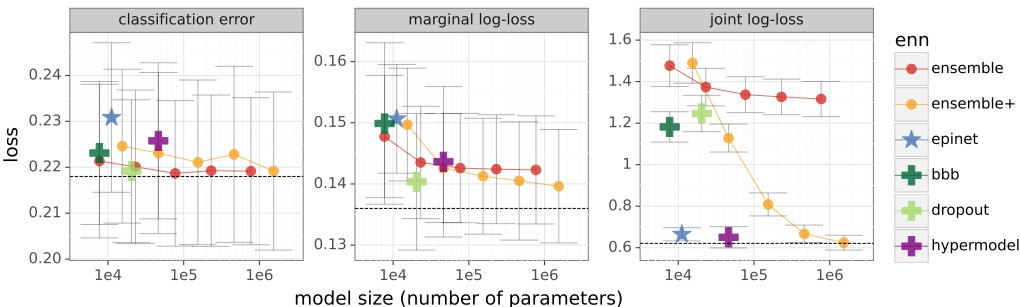

Figure 5: Quality of marginal and joint predictions across models on the Neural Testbed.

et al., 2009); qualitatively similar results for both CIFAR-10 and CIFAR-100 are presented in Appendix H. We compare our epinet agent against ensemble approaches as well as the *uncertainty baselines* of Nado et al. (2021). Even after tuning to optimize joint log-loss, none of these agents match epinet performance. We assess joint log-loss via dyadic sampling (Osband et al., 2022b), as explained in Appendix F.

For our experiments, we first train several baseline ResNet architectures on ImageNet. We train each of the ResNet-$L$ architectures for $L \in \{50, 101, 152, 200\}$ in the Jaxline framework (Babuschkin et al., 2020). We tune the learning rate, weight decay and temperature rescaling (Wenzel et al., 2020) on ResNet-50 and apply those settings to other ResNets. The ensemble agent only uses the ResNet-50 architecture. After tuning hyperparameters, we independently initialize and train 100 ResNet-50 models to serve as ensemble particles. These models are then used to form ensembles of sizes 1, 3, 10, 30, and 100.

The epinet takes a pretrained ResNet as the base network with frozen weights. We fix the index dimension $D_Z = 30$ and let the features $\phi$ be the last hidden layer of the ResNet. We use a 1-layer MLP with 50 hidden units for the learnable network (6). The fixed prior $\sigma^P$ consists of a network with the same architecture and initialization as $\sigma_\eta^L$, together with an ensemble of small random convolutional networks that directly take the image as inputs. We push details on hyperparameters and evaluation, with open-source code, to Appendix H.

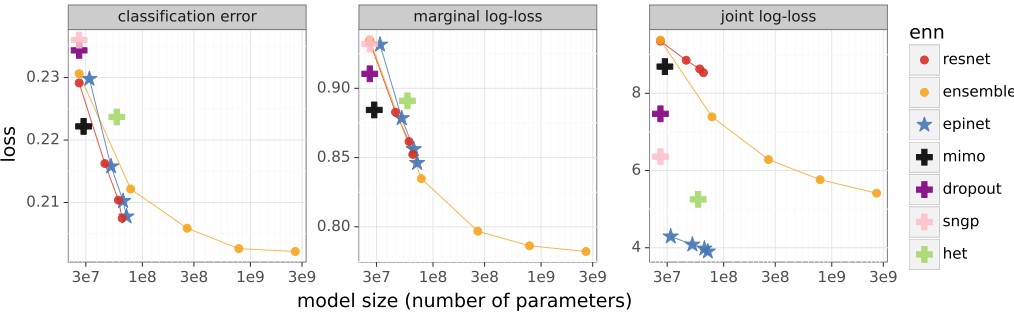

Figure 6: Marginal and joint predictions on ImageNet.

**Figure 2 presents our key result of this paper: relative to large ensembles, epinets greatly improve joint predictions at orders of magnitude lower compute cost.** The figure plots performance of three agents with respect to three notions of loss as a function of model size in the spirit of Kaplan et al. (2020). The first two plots assess marginal prediction performance in terms of classification error and log-loss. Performance of the ResNet scales similarly whether or not supplemented with the epinet. Performance of the ensemble does not scale as well as that of the ResNet. The third plot pertains to performance of joint predictions and exhibits dramatic benefits afforded by the epinet. While joint log-loss incurred by the ensemble agent improves with model size more so than the ResNet, the epinet agent outperforms both alternatives by an enormous margin.

**Figure 6 adds evaluation for the best single-model agents from *uncertainty baselines* (Nado et al., 2021): epinet also outperforms all of these methods**. We tuned each uncertainty baseline agent to minimize joint log-loss, subject to the constraint

that their marginal log-loss does not degrade relative to published numbers. We can see that the `epinet` offers substantial improvements in joint prediction compared to `sngp` (Liu et al., 2020), `dropout` (Gal and Ghahramani, 2016), `mimo` (Havasi et al., 2020) and `het` (Collier et al., 2020). As demonstrated in Appendix H, these qualitative observations remain unchanged under alternative measures of computational cost, such as FLOPs.

## 7  Conclusion

This paper introduces ENNs as a new interface for uncertainty modeling in deep learning. We do this to facilitate the design of new approaches and the evaluation of joint predictions. Unlike BNNs, which focus on inferring unknown network parameters, ENNs focus on uncertainty that matters in the *predictions*. The epinet is a novel ENN architecture that cannot be expressed as a BNN and significantly improves the tradeoffs in prediction quality and computation. For large models, the epinet enables joint predictions that outperform ensembles consisting of hundreds of particles at a computational cost only slightly more than one particle. Importantly, you can add an epinet to large pretrained models with modest incremental computation.

ENNs enable agents to know what they do not know, thereby unlocking intelligent decision making capabilities. Specifically, ENNs allow agents to employ sophisticated exploration schemes, such as information-directed sampling (Russo and Van Roy, 2014), when tackling complex online learning and reinforcement learning problems. A recent paper (Osband et al., 2023) has demonstrated efficacy of the epinet in several benchmark bandit and reinforcement learning problems. While ENNs can also be integrated into large language models, we defer this to future work.

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

# A  Open source code

Two related github repositories complement this paper:

1. `enn`: https://anonymous.4open.science/r/enn-55BC
2. `neural_testbed`: https://anonymous.4open.science/r/neural_testbed-8961

These libraries contain the code necessary to reproduce the key results in our paper, divided into repositories based on focus. Together with each repository, we include several 'tutorial colabs' – Jupyter notebooks that can be run in a browser without requiring any local installation. Each of these libraries is written in Python, and relies heavily on JAX for scientific computing (Bradbury et al., 2018). We view this open-source effort as a major contribution of our paper.

The first library, `enn`, focuses on the design of epistemic neural networks and their training. This includes all of our network definitions and loss functions. Our library is built around Haiku (Babuschkin et al., 2020). The library provides the basis for all of the computational work reported in this paper.

The second library, `neural_testbed`, was introduced as part of *The Neural Testbed* (Osband et al., 2022a). We add the `epinet` agent, suitable for comparison with the existing agent implementations in that library.

# B  From predictions to decisions

Suppose a neural network has been trained on a dataset $\mathcal{D}$ and, then, for a random input $x$ with label $y$ generates a distributional prediction $\hat{P}$. The expected marginal log loss is $-\mathbb{E}[\ln \hat{P}(y)|\mathcal{D}]$. Note that this expectation is over $x$, $y$, and $\hat{P}$. Results we establish point out that, while minimizing this produces optimal marginal predictions, that does not generally lead to performant downstream decisions.

To formulate a generic decision problem, consider a reward function $r$ that maps an action $a \in \mathcal{A}$ and $\tau$ labels $y_{1:\tau}$ to a reward $r(a, y_{1:\tau}) \in [0,1]$. Conditioned on training data $\mathcal{D}$ and random inputs $x_{1:\tau}$, an action $a$ generates expected reward

$$\mathbb{E}\left[r(a, y_{1:\tau})|\mathcal{D}, x_{1:\tau}\right] = \sum_{y_{1:\tau}} P_{1:\tau}(y_{1:\tau})r(a, y_{1:\tau}),$$

where $P_{1:\tau}(y_{1:\tau}) = \mathbb{P}(y_{1:\tau} = \cdot|\mathcal{D}, x_{1:\tau})$ is the posterior predictive distribution. An optimal action $a_*$ can be determined by maximizing this conditional expectation. If an approximation $\hat{P}_{1:\tau}$ is used instead of $P_{1:\tau}$ then the objective becomes $\sum_{y_{1:\tau}} \hat{P}_{1:\tau}(y_{1:\tau})r(a, y_{1:\tau})$.

Let $P_t = \mathbb{P}(y_t = \cdot|\mathcal{D}, x_t) = \sum_{y_{1:t-1}, y_{t+1,\tau}} P_{1:\tau}(y_{1:\tau})$ be the marginal posterior predictive. Let $\hat{P}_t = \sum_{y_{1:t-1}, y_{t+1,\tau}} \hat{P}_{1:\tau}(y_{1:\tau})$ be the approximate marginal. A joint prediction $\hat{P}_{1:\tau}$ minimizes marginal cross-entropy when $\hat{P}_t = P_t$ for $t = 1, \ldots, \tau$. The following result establishes that such predictions that minimize marginal log loss can result in decisions no better than random guesses. Note that when it is clear from context, we use notation for a set, like $\mathcal{A}$, to express cardinality of the set. The exchangeability requirement – that the distribution of any set of data pairs be exchangeable – ensures that we are in a standard supervised learning setting.

**Theorem 1.** [formal] *For all $\tau$, there exists an action set $\mathcal{A}$ with $|\mathcal{A}| = 2^\tau$, an exchangeable data distribution, and a reward function $r$ with range $[0,1]$ such that if $\hat{P}_{1:\tau}(y_{1:\tau}) = \prod_{t=1}^{\tau} \mathbb{P}(y_t|\mathcal{D}, x_{1:\tau})$ and $\hat{a} \sim \mathrm{unif}(\arg\max_{a\in\mathcal{A}} \sum_{y_{1:\tau}} \hat{P}_{1:\tau}(y_{1:\tau})r(a, y_{1:\tau}))$ then*

$$\max_{a\in\mathcal{A}} \mathbb{E}[r(a, y_{1:\tau})|\mathcal{D}, x_{1:\tau}] = 1 \qquad and \qquad \mathbb{E}[r(\hat{a}, y_{1:\tau})|\mathcal{D}, x_{1:\tau}] \le \mathbb{E}[r(\tilde{a}, y_{1:\tau})|\mathcal{D}, x_{1:\tau}] < 1,$$

*where, conditioned on $\mathcal{D}$ and $x_{1:\tau}$, $\tilde{a} \sim \mathrm{unif}(\mathcal{A})$.*

*Proof.* Let $\tau$ be even; extending our argument to odd $\tau$ is straightforward. Without loss of generality, let the input space be a singleton. Hence, $\mathbb{P}(y_{1:\tau} = \cdot|\mathcal{D}, x_{1:\tau}) = \mathbb{P}(y_{1:\tau} = \cdot|\mathcal{D})$.

Let labels $y_t$ be in $\{-1, 1\}$ so that $y_{1:\tau} \in \{-1, 1\}^\tau$. Let $\mathcal{A} = \{-1, 1\}^\tau$. Let $r(a, y_{1:\tau}) = 1 - (\sum_{t=1}^\tau a_t y_t / \tau)^2$. Let $\mathbb{P}(y_1 = \cdots = y_\tau | \mathcal{D}) = 1$ and, for all $t$, $\mathbb{P}(y_t = 1 | \mathcal{D}) = 1/2$. Clearly, this data distribution is exchangeable, with uniform marginals. Hence, for any $a \in \mathcal{A}$,

$$\sum_{y_{1:\tau}} \hat{P}_{1:\tau}(y_{1:\tau}) r(a, y_{1:\tau}) = \sum_{y_{1:\tau}} \hat{P}_{1:\tau}(y_{1:\tau}) \left(1 - \left(\frac{1}{\tau} \sum_{t=1}^\tau a_t y_t\right)^2\right)$$

$$= 1 - \sum_{y_{1:\tau}} \hat{P}_{1:\tau}(y_{1:\tau}) \left(\frac{1}{\tau} \sum_{t=1}^\tau y_t\right)^2$$

$$= 1 - \frac{1}{\tau^2} \left(\sum_{t=1}^\tau \sum_{y_{1:\tau}} \hat{P}_t(y_t) y_t^2\right)$$

$$= 1 - 1/\tau.$$

It follows that any action $\hat{a}$ maximizes $\sum_{y_{1:\tau}} \hat{P}_{1:\tau}(y_{1:\tau}) r(a, y_{1:\tau})$, and therefore, selecting randomly from this set results in expected reward $\mathbb{E}[r(\hat{a}, y_{1:\tau}) | \mathcal{D}, x_{1:\tau}] = 1 - 1/\tau$. Hence,

$$\mathbb{E}[r(\hat{a}, y_{1:\tau}) | \mathcal{D}, x_{1:\tau}] = \mathbb{E}[r(\tilde{a}, y_{1:\tau}) | \mathcal{D}, x_{1:\tau}]$$

$$= 1 - \mathbb{E}\left[\left(\frac{1}{\tau} \sum_{t=1}^\tau \tilde{a}_t y_t\right)^2 \middle| \mathcal{D}, x_{1:\tau}\right]$$

$$= 1 - \mathbb{E}\left[\left(\frac{1}{\tau} \sum_{t=1}^\tau \tilde{a}_t\right)^2\right]$$

$$= 1 - \frac{1}{\tau^2} \sum_{t=1}^\tau \mathbb{E}[\tilde{a}_t^2]$$

$$= 1 - 1/\tau.$$

For any action $a$ such that $\sum_{t=1}^\tau a_t = 0$,

$$\mathbb{E}[r(a, y_{1:\tau}) | \mathcal{D}, x_{1:\tau}] = \mathbb{P}(y_{1:\tau} = -\vec{1} | \mathcal{D}) \left(1 - \left(\frac{1}{\tau} \sum_{t=1}^\tau -a_t\right)^2\right)$$

$$+ \mathbb{P}(y_{1:\tau} = \vec{1} | \mathcal{D}) \left(1 - \left(\frac{1}{\tau} \sum_{t=1}^\tau a_t\right)^2\right)$$

$$= 1.$$

The result follows. $\qquad\square$

The first displayed equation of this theorem asserts that the optimal expected reward is one, while the second asserts that an action selected based on $\hat{P}_{1:\tau}$, which minimizes marginal log loss, does no better than an action chosen uniformly at random, which earns expected reward less than one.

While optimizing marginal predictions does not necessarily lead to performant downstream decisions, minimizing joint log loss does. Our next result formalizes this by bounding performance shortfall by a function of the KL-divergence:

$$\mathbf{d}_{\mathrm{KL}}(P_{1:\tau} \| \hat{P}_{1:\tau}) = \sum_{y_{1:\tau}} P_{1:\tau}(y_{1:\tau}) \ln P_{1:\tau}(y_{1:\tau}) - \sum_{y_{1:\tau}} P_{1:\tau}(y_{1:\tau}) \ln \hat{P}_{1:\tau}(y_{1:\tau}).$$

Only the final term depends on the prediction $\hat{P}$. Its conditional expectation $-\mathbb{E}[\ln \hat{P}(y_{1:\tau}) | \mathcal{D}]$ is the log loss. Hence, minimizing expected KL divergence is equivalent to minimizing log loss. This result follows almost immediately from Pinsker's and Jensen's inequalities. A proof can be found in (Wen et al., 2022).

**Theorem 2.** [formal] *For all data distributions, reward functions $r$ with range $[0,1]$, and actions $\hat{a} \in \arg\max_a \sum_{y_{1:\tau}} \hat{P}_{1:\tau}(y_{1:\tau})r(a, y_{1:\tau})$,*

$$\mathbb{E}[r(\hat{a}, y_{1:\tau})|\mathcal{D}, x_{1:\tau}] \geq \max_{a \in \mathcal{A}} \mathbb{E}[r(a, y_{1:\tau})|\mathcal{D}, x_{1:\tau}] - \sqrt{2\mathbb{E}[\mathbf{d}_{\mathrm{KL}}(P_{1:\tau}\|\hat{P}_{1:\tau})|\mathcal{D}, x_{1:\tau}]}.$$

The KL-divergence is the difference between the log loss attained by $\hat{P}_{1:\tau}$ and that attained by an optimal prediction $P_{1:\tau}$. The shortfall of action $\hat{a}$ is hence bounded by a measure of joint prediction error. If this error is small, $\hat{P}_{1:\tau}$ leads to good decisions regardless of the reward function.

## C   ENNs versus BNNs

In this appendix, we provide a proof of Theorem 3.

**Theorem 3.** *For all base networks $f$, any BNN defined with respect to $f$ can be expressed as an ENN defined with respect to $f$. However, there exists a base network $f$ and ENN defined with respect to $f$ that can not be expressed as a BNN defined with respect to $f$.*

*Proof.* Consider a BNN $(f, p)$. Without loss of generality, take the parameter space of $f$ to be $\mathbb{R}^d$. Let $P_Z$ be an absolutely continuous reference distribution over $\mathbb{R}^d$. Via Knothe-Rosenblatt rearrangement (Knothe, 1957; Rosenblatt, 1952), for each $\nu \in \mathbb{R}^d$, there exists a transport map from $P_Z$ to $p_\nu$. For each epistemic index $z \in \mathbb{R}^d$ and BNN parameter vector $\nu$, let $\hat{\theta}_{\nu,z}$ be the corresponding base network parameters generated by this transport map. Let $f'_\nu(x, z) = f_{\hat{\theta}_{\nu,z}}(x)$. It is easy to see that the ENN $(f', P_Z)$ is defined with respect to $f$ and expresses the BNN $(f, p)$.

For the second part of the theorem, consider a linear base network $f_\theta(x) = \theta^\top x$, a standard Gaussian epistemic index reference distribution, and an ENN

$$f'_{\theta'}(x, z) = f_\theta(x) + \theta'' z,$$

where $\theta' = (\theta, \theta'')$. Clearly, there are ENN parameters $\theta'$ and an epistemic index $z$ such that no $\hat{\theta}$ satisfies $f'_{\theta'}(x, z) = f_{\hat{\theta}}(x)$. If follows that no BNN defined with respect to $f$ can express the ENN. $\qquad\square$

## D   Bayesian linear regression

In this appendix, we provide a proof of Theorem 4. First, we state without proof a standard result on Bayesian linear regression (Minka, 2000).

**Lemma 1.** *Let $\mathcal{D} = \{(x_i, y_i, i)\}_{i=1}^N$ be generated by a linear-Gaussian model. Conditioned on $\mathcal{D}$, $\nu_*$ is Gaussian and*

$$\mathbb{E}[\nu_*|\mathcal{D}] = \left(\frac{1}{\sigma^2}\sum_{i=1}^N x_i x_i^\top + \frac{1}{\sigma_0^2}I\right)^{-1}\left(\frac{1}{\sigma^2}\sum_{i=1}^N x_i y_i\right), \quad \mathrm{Cov}[\nu_*|\mathcal{D}] = \left(\frac{1}{\sigma^2}\sum_{i=1}^N x_i x_i^\top + \frac{1}{\sigma_0^2}I\right)^{-1}. \tag{10}$$

Next, we prove a result on the near-orthogonality of vectors sampled uniformly from a unit sphere. Note that *a.s.* abbreviates *almost surely*.

**Lemma 2.** *Let $b_n$ and $c_n$ be independent vectors sampled uniformly from the $n$-dimensional unit sphere. Then,*

$$\lim_{n \to \infty} b_n^\top c_n \overset{a.s.}{=} 0.$$

*Proof.* Note that unit random vectors generated by sampling normal vectors from $N(0, I)$ and normalizing are uniformly distributed over the $\mathbb{R}^d$ unit sphere.

Let $\alpha_n$ and $\beta_n$ be independent samples from a chi-squared distribution with $n$ degrees of freedom. Let $u_n = \alpha_n^{1/2}b_n$ and $v_n = \beta_n^{1/2}c_n$. Clearly the distribution of both $u_n$ and $v_n$ is

isotropic, and since $\alpha_n$ and $\beta_n$ are distributed chi-squared with $n$ degrees of freedom (which is that of the length of $n$-dimensional standard normal), $u_n$ and $v_n$ are independent standard normal random vectors. Further, $b_n = u_n/\|u_n\|_2$ and $c_n = v_n/\|v_n\|_2$.

For any $n$,

$$
\begin{aligned}
b_n^\top c_n &= \frac{\sum_{i=1}^n u_i v_i}{\sqrt{\sum_{i=1}^n u_i^2 \sum_{i=1}^n v_i^2}} \\
&= \frac{\frac{\sum_{i=1}^n u_i v_i}{n}}{\sqrt{\frac{\sum_{i=1}^n u_i^2}{n} \frac{\sum_{i=1}^n v_i^2}{n}}}
\end{aligned}
$$

Since $\{u_i\}_{i=1}^n$ and $\{v_i\}_{i=1}^n$ are i.i.d $N(0,1)$, by the strong law of large numbers,

$$
\lim_{n\to\infty} \frac{\sum_{i=1}^n u_i v_i}{n} \overset{\text{a.s.}}{=} E[u_i v_i] = 0,
$$

$$
\lim_{n\to\infty} \frac{\sum_{i=1}^n u_i^2}{n} \overset{\text{a.s.}}{=} E[u_i^2] = 1,
$$

$$
\lim_{n\to\infty} \frac{\sum_{i=1}^n v_i^2}{n} \overset{\text{a.s.}}{=} E[v_i^2] = 1.
$$

Hence, by the continuous mapping theorem,

$$
\lim_{n\to\infty} b_n^\top c_n \overset{\text{a.s.}}{=} 0.
$$

$\square$

**Theorem 4.** *Let data $\mathcal{D} = \{(x_i, y_i, i)\}_{i=1}^N$ be generated by a linear-Gaussian model and $f$ be a linear-Gaussian epinet. Let $\hat\theta \in \arg\min_\theta \sum_{i=1}^N \int_z P_Z(dz) \ell_{\sigma,\lambda}^{\text{LSG}}(\theta, z, x_i, y_i, i)$ with parameter $\lambda = \sigma^2/(N\sigma_0^2)$. Then, conditioned on $(\mathcal{D}, c_{1:N}, P_0)$, $f_{\hat\theta}(\cdot, z)$ converges in distribution to $g_{\nu_*}$ as $D_Z$ grows, almost surely.*

*Proof.* The statement that, conditioned on $(\mathcal{D}, c_{1:N}, P_0)$, $f_{\hat\theta}(\cdot, z)$ converges in distribution to $g_{\nu_*}$ as $D_Z$ grows can be restated as

$$
\mathbb{P}\left(g_{\nu_*} \in F | \mathcal{D}\right) \overset{a.s.}{=} \lim_{D_Z \to \infty} \mathbb{P}(f_{\hat\theta}(\cdot, z) \in F | \mathcal{D}, c_{1:N}, P_0), \tag{11}
$$

for all measurable sets $F$ of functions mapping $\mathbb{R}^D$ to $\Re$.

Note that

$$
\hat\theta = (\hat\zeta, \hat\eta) \in \arg\min_{\zeta, \eta} \sum_{i=1}^N \int_z P_Z(dz)(f_{\zeta,\eta}(x_i, z) - y - \sigma c_i^T z)^2 + \frac{\sigma^2}{\sigma_0^2}\left(\|\zeta\|_2^2 + \|\eta\|_2^2\right).
$$

Since the optimization problem is strictly convex in $\theta$, $\hat\theta$ is the unique minimizer of this expression. Some simple algebra establishes that

$$
\hat\zeta = \hat\Sigma\left(\frac{1}{\sigma^2}\sum_{i=1}^N x_i y_i\right) \text{ and } \hat\eta = \left(\frac{1}{\sigma}\sum_{i=1}^N c_i x_i^\top + \frac{1}{\sigma_0} P_0\right)\hat\Sigma - P_0, \tag{12}
$$

$$
\text{where } \hat\Sigma = \left(\frac{1}{\sigma^2}\sum_{i=1}^N x_i x_i^\top + \frac{1}{\sigma_0^2} I\right)^{-1}.
$$

For any $x \in \mathbb{R}^D$ and $z \in \mathbb{R}^{D_Z}$, $f_{\hat\zeta, \hat\eta}(x, z) = \left(\hat\zeta + z^\top(\hat\eta + P_0)\right)^\top x$ and $g_{\nu_*}(x) = \nu_*^\top x$. Hence, (11) holds if and only if $\hat\zeta + z^\top(\eta + P_0)$ converges in distribution to $\nu_*$, conditioned on $(\mathcal{D}, \{c_i\}_{i=1}^N, P_0)$, almost surely. Since $\nu_*$ and $\hat\zeta + z^\top(\eta + P_0)$ are Gaussian, it is sufficient to show that

$$
\lim_{D_Z \to \infty} \hat\zeta = \mathbb{E}[\nu_* | \mathcal{D}] \quad \text{and} \quad \lim_{D_Z \to \infty} (\eta + P_0)^\top(\eta + P_0) = \text{Cov}(\nu_* | \mathcal{D}).
$$

Based on (10) and (12), $\hat{\eta} = \mathbb{E}[\nu_* | \mathcal{D}]$ for any $D_Z$. Hence, in order to prove the theorem, it is sufficient to show

$$\lim_{D_Z \to \infty} (\eta + P_0)^\top (\eta + P_0) = \text{Cov}(\nu_* | \mathcal{D}). \tag{13}$$

For any $D_Z$,

$$(\eta + P_0)^\top (\eta + P_0) = \hat{\Sigma} \left( \frac{1}{\sigma} \sum_{i=1}^N c_i x_i^\top + \frac{1}{\sigma_0} P_0 \right)^\top \left( \frac{1}{\sigma} \sum_{i=1}^N c_i x_i^\top + \frac{1}{\sigma_0} P_0 \right) \hat{\Sigma}$$

$$= \hat{\Sigma} \left( \frac{1}{\sigma^2} \sum_{i=1}^N x_i x_i^\top + \frac{1}{\sigma_0^2} I \right) \hat{\Sigma} + \hat{\Sigma} \left( \frac{1}{\sigma_0^2} \left( P_0^\top P_0 - I \right) \right) \hat{\Sigma}$$

$$+ \hat{\Sigma} \left( \frac{1}{\sigma^2} \sum_{i=1,j=1,i\neq j}^N c_i^\top c_j x_i x_j^\top + \frac{1}{\sigma \sigma_0} \sum_{i=1}^N (x_i c_i^\top P_0 + P_0^\top c_i x_i^\top) \right) \hat{\Sigma}$$

Recall that $\{c_i\}_{i=1}^N$ and columns of $P_0$ are all sampled i.i.d and uniformly from unit sphere in $\mathbb{R}^{D_Z}$, by Lemma 2,

$$\lim_{D_Z \to \infty} c_i^\top c_j \overset{\text{a.s.}}{=} 0 \forall i \neq j,$$

$$\lim_{D_Z \to \infty} P_0^\top c_i \overset{\text{a.s.}}{=} 0 \forall i$$

$$\lim_{D_Z \to \infty} P_0^\top P_0 \overset{\text{a.s.}}{=} I.$$

Hence,

$$\lim_{D_Z \to \infty} (\eta + P_0)^\top (\eta + P_0) \hat{\Sigma} \overset{\text{a.s.}}{=} \hat{\Sigma} \left( \frac{1}{\sigma^2} \sum_{i=1}^N x_i x_i^\top + \frac{1}{\sigma_0^2} I \right) = \hat{\Sigma}.$$

$\square$

## E   Didactic examples

To offer some intuition for how epinets work and what they accomplish, we present a simple example specialized to a linear base model. A linear base model produces an output $\mu_\zeta(x) = \zeta^T x$ given an input $x$ and model parameters $\zeta$. It is natural to add to this a linear epinet $\sigma_\eta(\phi_\zeta(x), z) = z^T \eta x$. The combined architecture is equivalent to a linear *hypermodel* (Dwaracherla et al., 2020). To see this, note that,

$$f_\theta(x, z) = \mu_\zeta(x) + \sigma_\eta(\phi_\zeta(x), z) = \zeta^T x + z^T \eta x = (\zeta + \eta^T z)^T x = \mu_{\zeta + \eta^T z}(x). \tag{14}$$

As such, properties of linear hypermodels, such as their ability to implement exact Bayesian linear regression, carry over to such epinets.

Figures 7 illustrates predictive uncertainty estimates produced by linear epinets. In this figure, posterior credible intervals of an epinet are compared against exact Bayesian inference. Data is generated by a one-dimensional linear regression model with a Gaussian prior distribution and Gaussian noise. The loss function for this epinet follows prior work by includes Gaussian bootstrapping in regression (Lu and Van Roy, 2017; Dwaracherla et al., 2020),

$$\ell(\theta, x_i, y_i, z) = (f_\theta(x_i, z) - y_i + \sigma c_i^T z)^2.$$

Here $c_i$ is a random signature drawn from the unit sphere in $D_Z$ generated independently for each training example $(x_i, y_i)$ and $\sigma$ is the scale of the additive bootstrap noise. This figure indicates that the epinet outputs well-calibrated marginal predictive distributions.

We next consider classification with a two-dimensional input and two classes. Data is generated by a standard logistic regression model with parameters drawn from a Gaussian prior. Figure 8 presents standard deviations of marginal predictive distributions across the input space. We supplement a standard logistic regression model with a linear epinet. The

plots compare results against those generated via SGMCMC, which we expect in this case to closely approximate exact Bayesian inference. While these figures bear qualitative similarities, significant differences arise because the linear epinet architecture imposes symmetries that are not respected by exact posterior distributions. In particular, this epinet can be thought of as representing parameter uncertainty as Gaussian. While our data generating process assumes a Gaussian prior distribution, the posterior distributions, which are conditioned on binary outcomes, are not Gaussian. More complex, nonlinear, epinets should be able to more accurately represent the posterior distribution over classifiers.

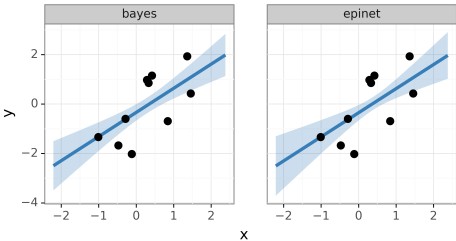

Figure 7: Epinet predictions in Gaussian linear regression.

Figure 8: Epinet predictions in Logistic regression.

## F  Dyadic sampling

To evaluate the quality of joint predictions, we sample batches of inputs $(x_1, \ldots, x_\tau)$ and assess log-loss with respect to corresponding labels $(y_1, \ldots, y_\tau)$. With a high-dimensional input space, labels of inputs sampled uniformly at random are typically nearly independent. Hence, a large batch size $\tau$ is required to distinguish joint predictions that effectively reflect interdependencies among labels. However, this becomes impractical because computational requirements for testing grow exponentially in $\tau$. Dyadic sampling serves as a practical heuristic that samples inputs more strategically so that effective agents are distinguished with a manageable batch size $\tau$ (Osband et al., 2022b) even when inputs are high-dimensional.

### F.1  Basic version

The basic version of dyadic sampling, for each batch, first samples two independent random anchor points, $\tilde{x}_1$ and $\tilde{x}_2$, from the input distribution. Then, to form a batch of size $\tau$, sample $\tau$ points independently and with equal probability from these two anchor points $\{\tilde{x}_1, \tilde{x}_2\}$. To assess an agent, its joint prediction of labels is evaluated for this batch of size $\tau$. Even with a moderate value of $\tau = 10$, a batch produced by dyadic sampling gives rise to labels that are likely to correlate – in particular, labels assigned to the same anchor point. Osband et al. (2022b) demonstrate that, across many problems, this sampling heuristic is effective in distinguishing the quality of joint predictions with modest computation.

On the Neural Testbed, the input distribution is standard normal. Thus, for each test batch, we sample anchor points $x_1, x_2 \sim N(0, I)$ and then re-sample $\tau = 10$ points from these two anchor points to form a batch.

On ImageNet, for faster evaluation, rather than sampling anchor points from the evaluation set, we split the evaluation set into batches of size 2. We then iterate over these batches of size 2, re-sample $\tau = 10$ points from each input pair, and evaluate the log-loss of an agent's joint predictions on these batches of size $\tau = 10$. Finally, we take the average of all the joint log-losses.

Figure 9(a) shows a few examples of these dyadic input batches of size $\tau = 10$. It may seem unsatisfactory that images are repeated exactly within a batch. Even though we view this metric as a unit test that an intelligent agent pass, it is conceivable to design a 'cheating' agent that takes advantage of this repeating structure. In the following section, we will consider a more robust version of dyadic sampling, where instead of repeating images exactly,

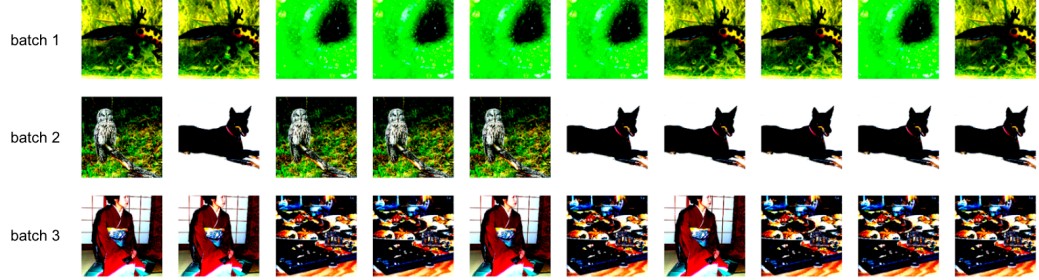

(a) Examples of input batches generated by basic dyadic sampling. Each row is a batch of size 10, on which the joint log-loss is evaluated.

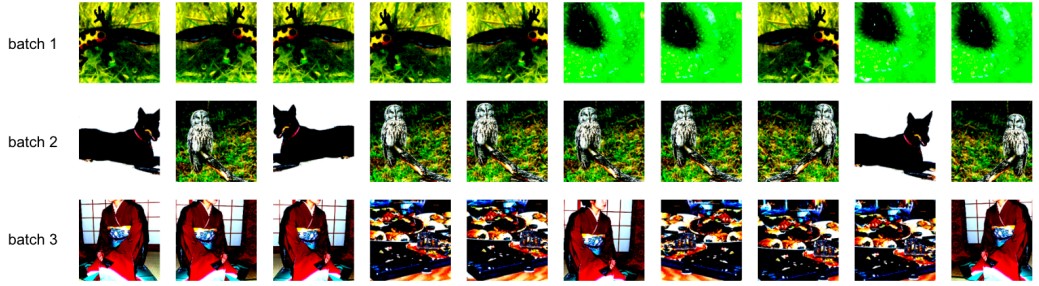

(b) Examples of input batches generated using augmented dyadic sampling. Compared to Figure 9(a), each image is randomly cropped and flipped, introducing more diversity to the batch.

Figure 9: Examples of input batches used for evaluating joint predictions. Figure 9(a) is generated through dyadic sampling and Figure 9(b) includes additional perturbations.

we perturb images using standard data augmentation techniques to introduce more diversity within a dyadic batch.

## F.2 Augmented dyadic sampling

In vanilla dyadic sampling, each dyadic batch has two unique elements (the anchor points), which are repeated multiple times within the batch. To make the metric more robust at distinguishing agents, we independently perturb each input in a dyadic batch using standard data augmentation techniques, so each input within a batch differs from others. For ImageNet, the perturbation takes the form of random cropping and flipping. We take the label of each perturbed image to be the label of its original image. Effective joint predictions indicate that images perturbed from the same anchor image are likely to be the same. Joint log-loss penalizes agents that do not recognize this. We call this sampling scheme *augmented dyadic sampling*. Figure 9(b) presents a few examples of augmented dyadic batches for Imagenet.

We evaluate our trained ResNet, ensemble, and epinet agents in Section 6 using augmented dyadic sampling, and we compare the joint log-loss with that obtained from basic dyadic sampling. In Figure 10, we see that the results from using these two sampling schemes are qualitatively similar. While the overall joint log-loss is higher for augmented dyadic sampling due to added perturbations, all agents benefit from increasing the model size. The joint log-loss of the ensemble agent improves more so than the ResNet agent with increasing model size. More importantly, the epinet agent outperforms both baselines by a huge margin under both sampling schemes. These results give us further confidence in the quality of joint predictions produced by the epinet agent.

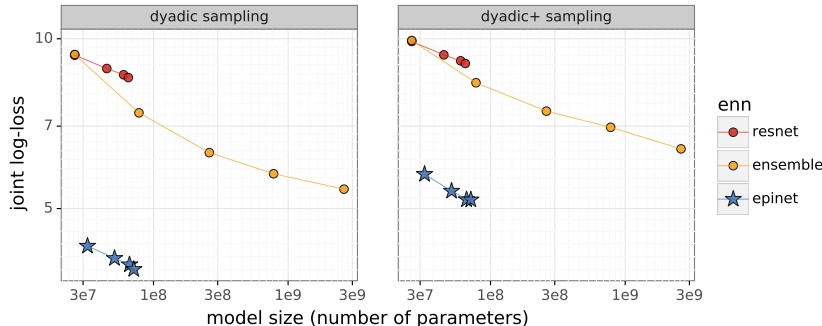

Figure 10: An agent's joint log-loss under dyadic (left) and augmented dyadic sampling (right).

# G Testbed experiments

This section provides details about the Neural Testbed experiments in Section 5. We begin with a review of the neural testbed as a benchmark problem, and the associated generative models. We then give an overview of the baseline agents we compare against in our evaluation. Next, we provide supplementary details for the hyperparameters and implementation details for the epinet agent as outlined in Section 5. Finally, we investigate the sensitivity our hyperparameter choices when evaluated across the testbed.

## G.1 Neural testbed

The Neural Testbed (Osband et al., 2022a) is a collection of neural-network-based, synthetic classification problems that evaluate the quality of an agent's predictive distributions. We make use of the open-source code at `https://github.com/deepmind/neural_testbed`. The Testbed problems use random 2-layer MLPs with width 50 to generate training and testing data. The specific version we test our agents on entails binary classification, input dimension $D \in \{2, 10, 100\}$, number of training samples $T = \lambda D$ for $\lambda \in \{1, 10, 100, 1000\}$, temperature $\rho \in \{0.01, 0.1, 0.5\}$ for controlling the signal-to-noise ratio, and 5 random seeds for generating different problems in each setting. The performance metrics are averaged across problems to give the final performance scores.

## G.2 Benchmark agents

We follow Osband et al. (2022a) and consider the benchmark agents as in Table 1. We use the open-source implementation and hyperparameter sweeps at `https://anonymous.4open.science/r/neural_testbed-8961/agents/factories`. According to Osband et al. (2022a), the benchmark agents are carefully tuned on the Testbed problems, so we do not further tune these agents.

## G.3 Epinet

We take the reference distribution of the epistemic index to be a standard Gaussian with dimension $D_Z = 8$. The base network $\mu_\zeta$ has the same architecture as the baseline `mlp` agent, which is a 2-layer MLP with ReLU activation and 50 units in each hidden layer. The learnable part of the epinet $\sigma_\eta^L$ takes $\phi_\zeta(x)$ and index $z$ as inputs, where $\phi_\zeta(x)$ is the concatenation of $x$ and the last-layer features of the base network. The learnable network has the form $\sigma_\eta^L(\phi_\zeta(x), z) = g_\eta([\phi_\zeta(x), z])^T z$ where $[\phi_\zeta(x), z]$ is the concatenation of $\phi_\zeta(x)$ and $z$, and $g_\eta(\cdot)$ is a 2-layer MLP with hidden width 15, ReLU activation, and outputs in $\mathbb{R}^{D_Z \times C}$ for number of classes $C = 2$.

For the fixed prior $\sigma^P$, we consider an ensemble of $D_Z$ networks. Each member of the ensemble is a small MLP with 2 hidden layers and 5 units in each hidden layer. Each MLP takes $x$ as input and returns logits for the two classes. Let $p^i(x) \in \mathbb{R}^C$ denote the output of the $i^{\text{th}}$ member of the ensemble. We combine the outputs of the ensemble members by taking the weighted sum, $\sum_{i=1}^{D_Z} p^i(x) z_i$ and multiplying the sum by a tunable scaling factor $\alpha$. Thus, we can write the prior function $\sigma^P(\phi_\zeta(x), z) = \alpha \sum_{i=1}^{D_Z} p^i(x) z_i$.

We combine the base network and the epinet by adding their outputs and applying a stop-gradient operation on $\phi_\zeta(x)$,

$$f_\theta(x, z) = \mu_\zeta(x) + \sigma_\eta^L(\text{sg}[\phi_\zeta(x)], z) + \sigma^P(\text{sg}[\phi_\zeta(x)], z),$$

where $\theta = (\zeta, \eta)$ denotes all the parameters of the base net and epinet. We train the parameters $\zeta$ and $\eta$ jointly. The training loss takes the form as specified in (7), where we use log loss for the data loss and ridge regularization. We update $\theta$ using Algorithm 1, with a batch size of 100 and number of epistemic index samples equal to the index dimension. We use Adam optimizer with learning rate `1e-3`. The $L2$ weight decay and prior scaling factor $\alpha$ are roughly adjusted for different problem settings, taking in account the number of training samples and SNR.

Our implementation of the epinet agent can be found under the path `/agents/factories/epinet.py` in the anonymized neural testbed github.

### G.4   Ablation studies

We run ablation experiments on the epinet agent that is trained simultaneously along with the base network. We sweep over various values for the index dimension, the number of hidden layers in the trainable epinet, the width of hidden layers of epinet, the ensemble prior's prior scale, the width of the hidden layers in the prior network, and $L2$ weight decay. We keep all other hyperparameters fixed to the open-sourced default configuration while sweeping over one hyperparameter.

Our results are summarized in Figure 11. We see that a larger index dimension improves both joint and marginal kl estimates. The epinet performance is not sensitive to the number of hidden layers. We suspect that this is due to similar number of parameters across epinets with different number of hidden layers. We observe that epinet performance is not sensitive to width of the epinet hidden layers once the width is large enough. Performs of epinet seems sensitive to the prior scales. Smaller prior scale leads to better marginal kl, too small or too large prior scale degrades the joint kl. Increasing the width of the models in the ensemble prior network improves the epinet performance. However, the improvement seems marginal after a point. The epinet seems to be sensitive to $L2$ weight decay. A very small or large weight decay degrades the performance of epinet.

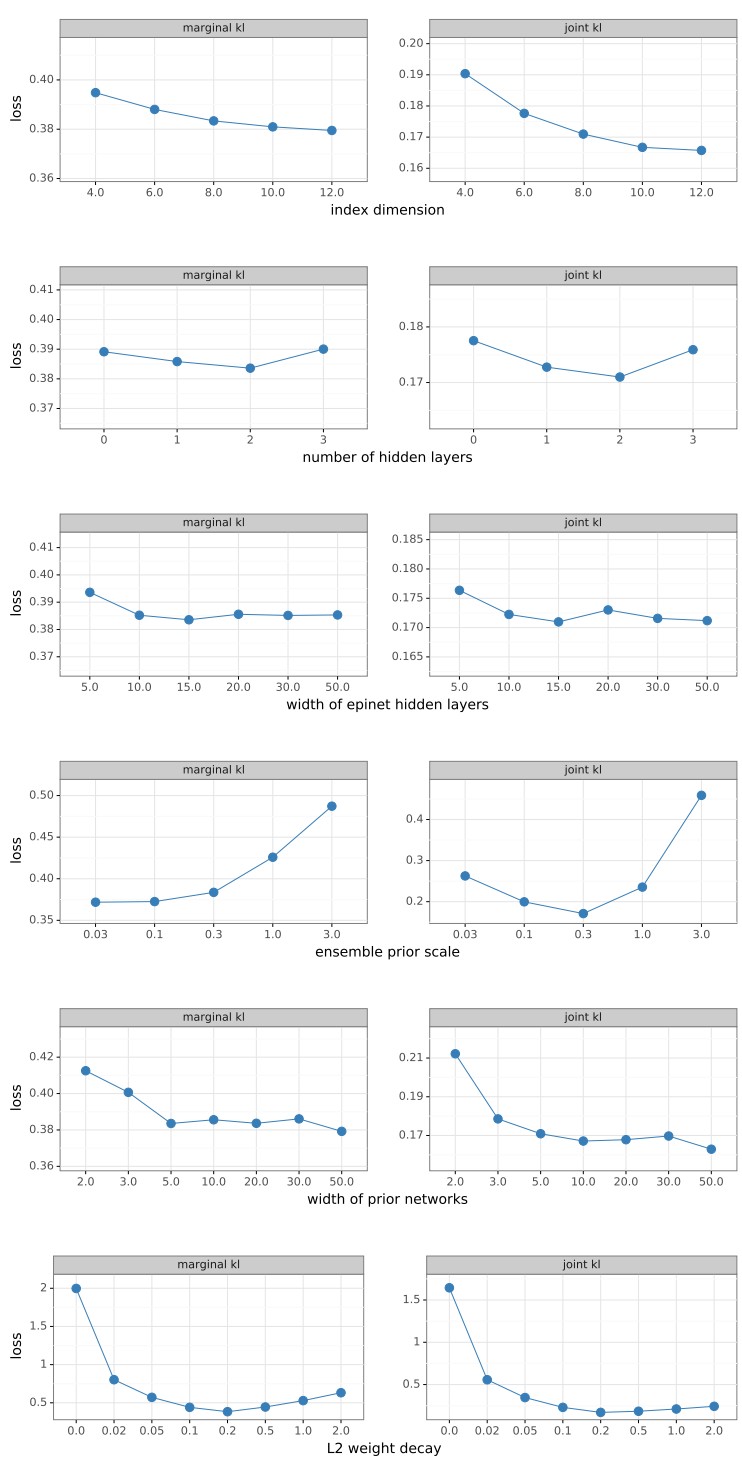

Figure 11: Ablation studies of epinet with 2-layer mlp base model on the neural testbed.

# H  Image classification

This section gives an overview of our experiments on image classification problems outlined in Section 6. We begin with a review of the hyperparameter choices and design details for the agents as implemented in our experiments. Then, we include a comparison of these agents to benchmark implementations in the field as embodied by 'uncertainty baselines' (Nado et al., 2021). Next, we present results for an evaluation on both CIFAR-10 and CIFAR-100, and find that the overall results match that on ImageNet. We complement these results with an analysis of the computational cost in terms of FLOPs as well as memory on modern TPU architectures. Finally, we perform a suite of ablations to investigate the sensitivity of our results across ImageNet.

## H.1  Epinet details

We train one epinet for each ResNet-$L$ baseline for $L \in \{50, 101, 152, 200\}$. The ResNet baselines are open sourced in the ENN library under the path `/networks/resnet/`, and the checkpoints are available under `/checkpoints/imagenet.py`. For each epinet agent, we take the pre-trained ResNet as the base network. We do not update the base network during epinet training. The epinet network architecture together with checkpoint weights can be found in the ENN library under the paths `/networks/epinet/` and `/checkpoints/imagenet.py`.

As discussed in Section 6, we choose the reference distribution of the epistemic index to be a standard Gaussian with dimension $D_Z = 30$. The input to the learnable part of the epinet $\sigma_\eta^L$ includes the last-layer features of the base ResNet and the epistemic index $z$. Let $C$ denote the number of classes. For last-layer features $\phi$ and epistemic index $z$, the learnable network takes the form $\sigma_\eta^L(\phi, z) = g_\eta([\phi, z])^T z$, where $[\phi, z]$ is the concatenation of $\phi$ and $z$, and $g_\eta(\cdot)$ is a 1-layer MLP with 50 hidden units, ReLU activation, and output $\in \mathbb{R}^{D_Z \times C}$.

The fixed prior $\sigma^P$ is made up of two components. The outputs of the two parts are summed together to produce the prior output. In general, we could have tunable scaling factors (which we refer to as 'prior scales' in the ablation studies) for the output of each component before we add them together. However, for ImageNet, we find that scaling factors of 1 already work well. The first component of the prior is a network with the same architecture and initialization as the learnable network $\sigma_\eta^L$. The second component is an ensemble of small convolutional networks that act directly on the input images. The number of networks in the ensemble is equal to the index dimension. Each convolutional network has the number of channels $(4, 8, 8)$, kernel shapes $(10 \times 10, 10 \times 10, 3 \times 3)$, and strides $(5, 5, 2)$. The outputs are flattened and taken through a linear layer to give a vector of dimension $C$. For input image $x$, let $p^i(x) \in \mathbb{R}^C$ denote the output of the $i^{\text{th}}$ member of the ensemble. We combine the outputs of the ensemble members by taking the weighted sum $\sum_{i=1}^{D_Z} p^i(x) z_i$.

The ResNet baselines are trained using log loss and ridge regularization. We optimize using SGD with a learning rate 0.1, a cosine learning rate decay schecule, and Nesterov momentum. We apply $L2$ weight decay of strength `1e-4`. We also incorporate label smoothing into the loss, where instead of one-hot labels, the incorrect classes receive a small weight of $0.1/C$. We train the ResNet agents for 90 epochs on $4 \times 4$ TPUs with a per-device batch size of 128.

We train the epinet using loss of the form (7) with log loss with ridge regularization. Similar to ResNet training, we apply $L2$ weight decay of strength `1e-4` and incorporate label smoothing into the loss. We draw 5 epistemic index samples for each gradient step. We optimize the loss using SGD with a learning rate 0.1, Nesterov momentum and decay 0.9. The epinet is trained on the same hardware with the same batch size for 9 epochs.

## H.2  Uncertainty baselines

In this section we compare our results to the open source 'uncertainty baselines', which provides a reference implementation for much work on uncertainty estimation in Bayesian deep learning (Nado et al., 2021). As part of our development, we upstream an optimized method for calculating joint log-loss, and contribute this to the community. We benchmark a few popular approaches to uncertainty estimation in terms of both marginal and joint predictions, and compare their results to ours. At a high level, our results mirror our own

results of Figure 2. After tuning, most of the agents appear on a roughly similar tradeoff in terms of marginal quality. However, the approachs are widely separated in terms of their quality on *joint* prediction, and here epinet performs much better than the alternatives.

Figure 6 repeats Figure 2 but adding a few new agents from uncertainty baselines, which we

- `mimo`: Multi-input Mulit-output ensemble (Havasi et al., 2020). This method appears to perform better than a baseline resnet in terms of marginal statistics, but provide no additional benefit to modeling joints. Note that the independent product of better marginals automatically does improve joints.
- `dropout`: Dropout as posterior approximation (Srivastava et al., 2014; Gal and Ghahramani, 2016). This approach seems to provide a slight improvement in marginal log-loss and a noticeable improvement in joint log-loss. However although dropout has a low computational cost in terms of *parameters*, these results required 10 forward passes of the network, and so actually *underperfom* relative to an ensemble of similar inference cost (see Appendix H.4).
- `sngp`: Spectral-normalized Neural Gaussian Process (Liu et al., 2020). For this agent we introduced an additional temperature parameter rescaling logit samples, which we found was able to significantly improve joint log-loss without degrading marginal prediction quality. However, even with this tuning the quality of joint predictions cannot match ensemble of size 10. The classification accuracy of SNGP also appears to be significantly worse than other approaches benchmarked here.
- `het`: Heteroscedastic loss (Collier et al., 2020, 2021). This approach performs well in terms of joint log-loss, and achieves performance close to the ensemble of size=100. Although the results are still significantly worse than those of our `epinet`, it is interesting to note that the functional form of the resultant `heteroscedastic` agent can actually be written as a particular form of epinet. In future work, we would like to understand better the commonalities between these two approaches, and see if/when the algorithms can borrow from each others' strengths.

### H.3 CIFAR-10 and CIFAR-100

In this section we reproduce a similar analysis to Section 6 applied to the CIFAR-10 and CIFAR-100 datasets. We find that, at a high level, our results mirror those when applying ResNet to ImageNet. In particular, we are able to produce results similar to Figure 2 for both CIFAR-10 and CIFAR-100. Relative to large ensembles, epinets greatly improve joint predictions at orders of magnitude lower computational cost.

For the experiments in this section, we mirror the ImageNet experiments but using the smaller ResNet architectures. The ResNet baselines are open sourced in the ENN library under the path `/networks/resnet/`, and the checkpoints are available under `/checkpoints/cifar10.py` and `/checkpoints/cifar100.py`. In particular, we tune ResNet-$L$ for $L \in \{18, 32, 44, 56, 110\}$ over learning rate and weight decay. We did not include temperature rescaling in these sweeps, although this could further improve performance for all agents. After tuning hyperparameters we independently initialize and train 100 ResNet-18 models to serve as ensemble particles. These models are then used to form ensembles of size 1, 3, 10, 30, and 100.

For the epinet agent we take the pretrained ResNet as the base network and fix its weights. We then follow the same methodology as for ImageNet, described in Appendix H.1, but with a slightly smaller network. We use index dimension $D_Z = 20$ and alter the convolutional prior to have channels $(4, 8, 4)$ each with kernel size $5 \times 5$ and stride 2 on account of the smaller image sizes in CIFAR-10 and CIFAR-100 (Krizhevsky, 2009).

The epinet network architecture together with checkpoint weights can be found in the ENN library under the paths `/networks/epinet/`, `/checkpoints/cifar10.py`, and `/checkpoints/cifar100.py`.

Figures 12 and 13 reproduce our scaling results for ImageNet when applied to these other datasets. At a high level, the key observations remain unchanged across datasets. We see that across all statistical losses the larger models generally perform better. When looking

at classification and marginal log-loss, epinets do not offer any particular advantage over baseline ResNets. However, when we look at the *joint* log-loss we can see that epinets offer huge improvements in performance, even when measured against very large ensembles.

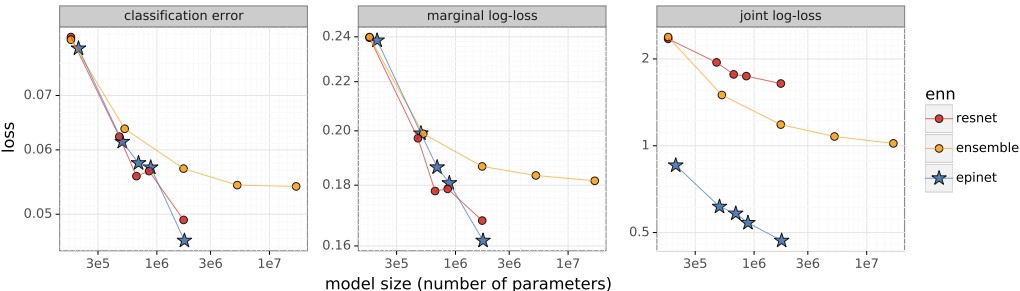

Figure 12: Quality of marginal and joint predictions across models on CIFAR-10.

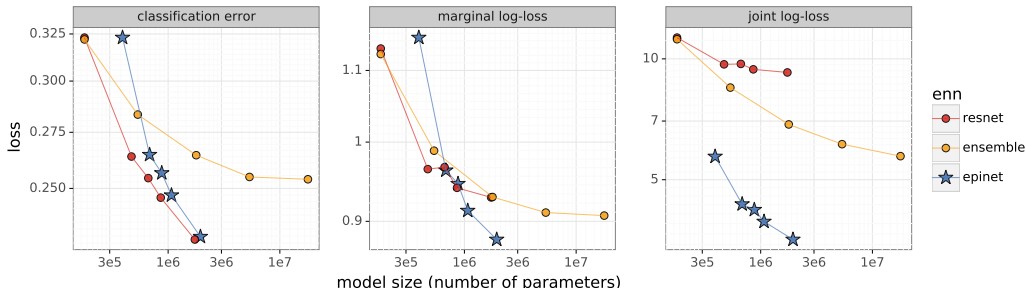

Figure 13: Quality of marginal and joint predictions across models on CIFAR-100.

## H.4   Computational cost

This paper highlights a key tension in neural network development: balancing statistical loss against computational cost. Our main results on ImageNet (Figure 2) use 'memory' as a proxy for computational cost in large deep neural networks, for which this is often a hardware bottleneck (Kaplan et al., 2020). However, in the case of these models, the results are similar for many alternative measures.

Figure 14 reproduces the results of Figure 2 but with computational cost measured in total floating point operations at inference. This plot includes the total costs of one thousand forwards of the epinet. Even with these additional FLOPs, the overall cost of each epinet is still less than 50% of the total network, and the outperformance of the epinet in terms of joint log-loss is still remarkable. Further, on large modern TPU architectures these epinet operations can often be performed in parallel. This means that, in some cases, these extra FLOPs may require no extra *time* to forward on the device.

The results of Figure 14 hide an extra hyperparameter in epinet development: the number of independent samples of the index $z$, which we will call $M$. All of the results presented in this paper focus on the case $M = 1000$, however an agent designer may choose to vary this depending on their tradeoff between statistical loss and computational cost. Figure 15 shows this empirical tradeoff over $M \in \{10, 30, 100, 300, 1000, 3000\}$ for each of the resnet variants. Once again, we see that in terms of *marginal* statistics, there is really no benefit to using an epinet. However, once you look at joint log-loss even a small number of epinet samples improves over the ResNet. Interestingly, these results continue to improve for $M > 1000$ but at a higher computational cost.

## H.5   Epinet ablations

We run ablation experiments on the epinet agent that builds on the pre-trained ResNet-50 base network. We sweep over various values for the index dimension, the number of hidden

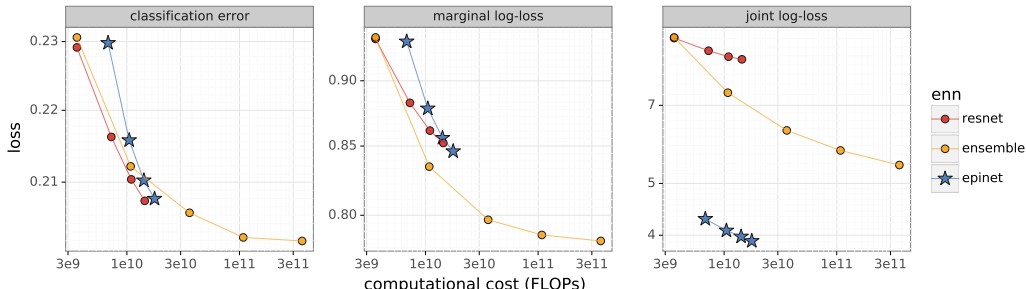

Figure 14: Quality of marginal and joint predictions across models on ImageNet. Reproduces the results of Figure 2 but using inference FLOPs as measure of computation.

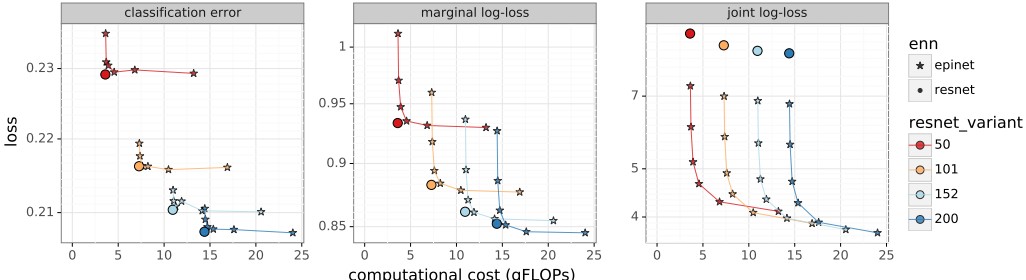

Figure 15: Comparing base ResNet against epinet for differing numbers of sampled indices $z$.

layers in the trainable epinet, the prior scale for the matched epinet prior, the prior scale for the ensemble of convolutional networks, $L2$ weight decay, the number of index samples drawn for each gradient step, label smoothing, and temperature re-scaling post-training. We keep all other hyperparameters fixed to the open-sourced default configuration while sweeping over one hyperparameter.

Our results are summarized in Figure 16. We see that a larger index dimension improves the joint loss-loss, but does not necessarily improve the marginal log-loss and classification error. Adding another hidden layer to the trainable epinet makes the marginal and joint los-loss worse, but we suspect that the performance could be improved with more tuning. The epinet performance seems sensitive to the prior scales. In the third and fourth rows, we see that the performance degrades quickly when the prior scales become too large. The epinet seems relatively robust to different values of $L2$ weight decay. A large weight decay improves the joint log-loss but slightly worsens the marginal log-loss. The number of index samples and degree of label smoothing do not seem to affect the performance of epinet. Interestingly, we find that using a cold temperature $< 1$ to re-scale the ENN output logits post-training improves the agent's performance during evaluation (Wenzel et al., 2020). The improvement is most dramatic in the joint log-loss.

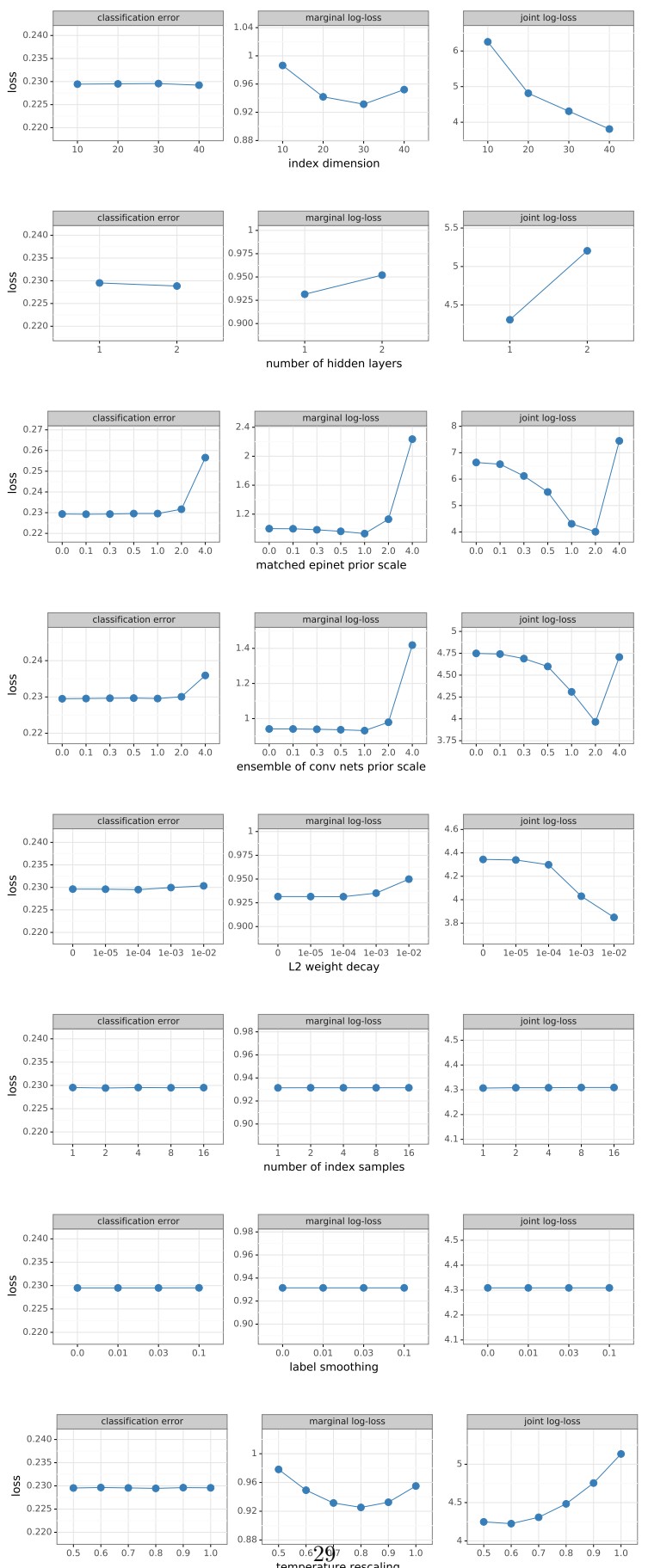

Figure 16: Ablation studies of epinet with ResNet-50 base model on ImageNet.