# OpenReview forum: "Epistemic Neural Networks"
_NeurIPS.cc/2023/Conference — NeurIPS 2023 spotlight_

### Official Review · Reviewer_tvti · 2023-06-24

**Soundness:** 2 fair
**Presentation:** 1 poor
**Contribution:** 3 good
**Rating:** 6
**Confidence:** 4

**Summary:**

This paper introduces *Epistemic Neural Networks (ENN)* as a general approach for uncertainty estimation for deep learning, and then the *epinet*, which is a particular ENN instantiation. The paper argues that *joint* predictions are essential for sequential decision making problem, and shows *epinets* offer similar marginal performance but better joint predictive performance than conventional BNNs/DNNs.

**Strengths:**

1. Interesting ideas, particularly around setting aside getting good weight-space posteriors and instead focusing on getting good predictive performance (whether that be marginal or joint predictions). I think this is good, and is of interest to the community.

2. Solid experimental results, especially around joint predictions. Nice job.

**Weaknesses:**

Although I think the paper is of interest to the community, I think the paper could be substantially improved. I have some concerns about the presentation and writing of the paper that limit how excited I am about this work being published at NeurIPS. On balance, I would like to see the suggestions I've made below incorporated into the writing of the text, and this would enable me to increase my score.

# 1. Poor Technical Presentation, Writing, Technical Soundness
I have a number of specific concerns here. Some of these might be seen as nit-picky, I admit, but I remain concerned here.
1. L7-9. "With an epinet, CNNs outperform large ensembles of hundred of more particles, and use orders of magnitude less compute." This claim should be qualified because it only applies robustly to joint-predictions. This applies elsewhere in the paper.

2. L31-32 "These approaches can almost match the posterior distribution." I think citing Welling and Teh for this doesn't make sense, and I don't think this is true. See, for example, the analysis in [1].

3. L36-37. "Practical large scale implementations are often limited to ten or fewer [particles] due to computational reasons". The way you've defined "ensemble based BNNs", sampling based inference or even VI when sampling from the approximate posterior counts as an ensemble. And some approaches use more than 10 samples, e.g., sampling 30+ particles from an approximate posterior should be fine.

4. In a number of places in the paper, intuition could be provided that would allow the reader to understand the points you are making more clearly. e.g., L50-51. "All BNNs are ENNs, but there are useful ENNs such as the epinet that are not BNNs". Explain why please. ENNs have the extra index, which mean that they have a wider class of functions (as far as I can tell), but I'm not sure this is right. L215-216 "with this stop gradient, training dynamics more reliably produce models that perform out ot sample". Please provide more intuition.

5. The writing in Section 3 only addresses classification problems. It should be more general.

6. The notation in 3.2 is confusing. Both $\nu$ and $\theta$ correspond to parameters.

7. It is worth noting that recent work argues that BNNs do not actually need to maintain distributions over all of their parameters for good predictive performance [1]. This is important because it's mentioned in the paper that BNNs learn distributions over all parameters, but this is not completely true, some BNNs do not do this. This characterisation seems wrong.

8. L163: "A BNN is specified by a pair: a base network $f$ and a parameterised sampling distribution $p$." I don't think this is quite right. I think a BNN is specified by a likelihood and prior, which defines a posterior distribution, at least technically speaking. Just because we have a distribution over network parameters does not mean that the network is a BNN. That might be contentious, but the sampling distribution certainly does not need to be parametric! For example, HMC has no parametric sampling distribution!

9. Eq (3): it seems to be the definition should integrate over $z$ and examine the distribution over outputs when doing that. The actual equation currently seems to be a stronger condition: correct joint predictions for every value of $z$. Am I missing something here?

10. Theorem 3. "any BNN defined with respect to $f$ can be expressed as an ENN defined with respect to $f$". I find this is a bit confused. Are there no technical conditions here? If there are technical conditions, please improve them. For instance, if there is a BNN with 10 million parameters, how do we represent this with an ENN that has one dimensional $Z$.

11. Similarly, notation in 4.1 is a bit confusing, with different symbols used for different parameters etc.

12. I think it is incredibly confusing that the paper introduces ENNs and epinets, but that these are not the same thing! I would suggest changing the name of one of them, probably the epinet.

# 2. Evaluations
1. What does the reduction in joint-log loss actually correspond to here? Does it yield better predictions in sequential decision making problems? Can you test this? For example, on an active learning or bandit problem? I appreciate this might be a tough ask, but I think given that decision making problems are a central focus for this work, this would be a substantial improvement to the submission. I'd also like to see the performance of an ENN on a 1D regression problem to understand whether ENNs offer coherent updates when observing data, since BNNs (ideally) would offer this. This could be done by retraining, or by just updating the posterior over the epistemic index.

2. I'd like to see results on ImageNet-C i.e., OOD robustness. BNNs and other approaches are often used here. I'm curious to see if the epinet helps here. Maybe this is what you are doing already, it wasn't clear to me reading the main paper alone, and it should be.

3. Are there error bars on Figure 2? It seems not? Please add them, if possible!

4. It is not clear what the reduction in joint log-loss corresponds to, the number is hard to interpret or understand. As a reader, I cannot understand how significant this is. Benchmarking on a sequential decision making problem and demonstrating improved decision making there would be a much more significant result. I appreciate there is theory here, but I'm more interested in the practical implications of the work.

5. Ideally, there would be examples of networks having good marginal predictions but poor joint predictions that lead to problems in sequential decision making problems.

[1] Sharma, Mrinank, et al. "Do Bayesian Neural Networks Need To Be Fully Stochastic?." International Conference on Artificial Intelligence and Statistics. PMLR, 2023.

**Questions:**

1. L42-43 "evaluating each model on out-of-sample data", is this OOD data e.g., from ImageNet-C, or just the normal test set? Please specify?

2. How would one actually use an ENN on sequential decision making problems or active learning problems? Update the epinet parameters only? Retrain the whole thing? Perform inference over Z? Given that this is the aim of the paper, this is an important topic to discuss in the main text.

3. Is the Glorot initialisation used for the MLP important?

**Limitations:**

Looks good to me.

---

> ### Author Rebuttal · Authors · 2023-08-10
>
> We thank the reviewer for carefully reviewing our paper and writing detailed reviews. Our responses are as follows. Due to the length limit of the rebuttal, we will omit parts of the questions and provide concise answers.
>
> **Regarding your questions:**
>
> **1. L42-43 "evaluating each model on out-of-sample data", is this OOD data e.g., from ImageNet-C, or just the normal test set? Please specify?**
>
> This is a typical test dataset. We will clarify.
>
> **2. How would one actually use an ENN on sequential decision making problems or active learning problems? Update the epinet parameters only? Retrain the whole thing? Perform inference over Z? Given that this is the aim of the paper, this is an important topic to discuss in the main text.**
>
> Thanks. In case of sequential decision making, on obtaining new data, both base model and epinet needs to be trained (as the data provides information for both base model and epinet). This can be done either concurrently or by first training the base network and then training the epinet in an iterative manner. We believe that both of these should be equivalent.
>
> **3. Is the Glorot initialisation used for the MLP important?**
>
> We think that Glorot initialisation would lead to outputs roughly in N(0,1) when inputs are from N(0, I). This helps us by not having to tune the scale for different depths and widths of the neural networks. But techniques presented in our paper should also work for other initialization schemes.
>
> **Regarding your comments on technical presentation, writing, and technical soundness:**
>
> **L7-9. "With an epinet, CNNs ... This applies elsewhere in the paper.**
>
> We will polish the wording in the revision.
>
> **L31-32 "These approaches can almost ... See, for example, the analysis in [1].**
>
> Thanks for pointing us to [1], we were not aware of this recent paper. We will make modifications accordingly.
>
> **L36-37. "Practical large scale ... e.g., sampling 30+ particles from an approximate posterior should be fine.**
>
> When we refer to “ensemble”, we just mean an ensemble of particles, not more general sampling based inference. We will change “ten or fewer particles” to “at most tens of particles”.
>
> **In a number of places in the paper ... Please provide more intuition.**
>
> Thanks for the comments! We will provide more intuitions in the revision to further improve the writing.
>
> **The writing in Section 3 only addresses classification problems. It should be more general.**
>
> We fully agree that it can be more general, in particular, it can also include the problem formulations for regression problems. We only include the classification formulation since we would like to keep the paper short and simpler.
>
> **The notation in 3.2 is confusing. Both \nu and \theta correspond to parameters.**
>
> Note that \theta corresponds to the parameters of the neural network, and \nu corresponds to the parameters of the sampling distribution.
>
> **It is worth noting that recent work ... some BNNs do not do this. This characterisation seems wrong.**
>
> Thanks for pointing us to this very recent work, we were not aware of it. We would like to clarify that we just would like to give a high-level overview of how classical BNNs model uncertainty. We will change the wording accordingly.
>
> **L163: "A BNN is specified by a pair... For example, HMC has no parametric sampling distribution!**
>
> We will clarify in the revision that the sampling distribution is supposed to be an approximation of the posterior distribution.
>
> **Eq (3): it seems to be the definition ... Am I missing something here?**
>
> Note that `z` is a random variable, so in Equation (3), we imply that the distribution of a BNN can be matched by the distribution generated by an ENN.
>
> **Theorem 3. "any BNN defined with respect to `f` can be expressed ... how do we represent this with an ENN that has one dimensional `Z`.**
>
> Theorem 3 doesn’t require any additional technical conditions. This can be observed from the proof of Theorem 3 in Appendix C.
>
> **Similarly, notation in 4.1 is a bit confusing, with different symbols used for different parameters etc.**
>
> We used different symbols to distinguish different parameters to avoid confusion.
>
> **I think it is incredibly confusing that the paper ..., probably the epinet.**
>
> ENNs are a more general framework. Epinet is one specific kind of ENN. For example, ENNs are to NNs as Epinet is to an MLP.
>
> **Regarding your comments on evaluations:**
>
> **What does the reduction in joint-log loss actually correspond to here? ...  or by just updating the posterior over the epistemic index.**
>
> Thanks for the great suggestion. We will include these results with 1D regression in the paper.
>
> **I'd like to see results on ImageNet-C i.e., OOD robustness. ..., it wasn't clear to me reading the main paper alone, and it should be.**
>
> Thanks for this suggestion! We will examine performance on Imagenet-C and other OOD datasets in our future work.
>
> **Are there error bars on Figure 2? It seems not? Please add them, if possible!**
>
> We weren't able to add the error bars because of the compute requirements for some of the agents (like ensembles), but we will try to include them in the final version.
>
> **It is not clear what the reduction in joint log-loss corresponds to, ... but I'm more interested in the practical implications of the work.**
>
> We would like to point the reviewer to a recent paper [1] that examines performance of epinet, in both the neural bandit problems and reinforcement learning problems.
>
> [1]  “Approximate Thompson sampling via Epistemic Neural Networks” https://openreview.net/pdf?id=xampQmrqD8U
>
> **Ideally, there would be examples of networks having good marginal predictions but poor joint predictions that lead to problems in sequential decision making problems.**
>
> Based on the paper [1] discussed above, it is indeed the case that the  agents which have good marginal predictions and poor joint predictions  perform poorly in sequential decision problems.

---

> > ### Comment · Reviewer_tvti · 2023-08-13
> >
> > Thanks. I raise my score to weak accept. I would like to re-iterate that I think the work is good but it needs some polish and improvement to writing. I would ask the authors of the papers to revise and improve the writing when submitting the camera ready.

---

> > > ### Author Response · Authors · 2023-08-15
> > >
> > > Dear reviewer, thanks for going through our rebuttal and increasing the score. We would also like to thank you for your constructive feedback and for engaging with us.

---

### Official Review · Reviewer_zTnm · 2023-07-07

**Soundness:** 3 good
**Presentation:** 3 good
**Contribution:** 3 good
**Rating:** 6
**Confidence:** 3

**Summary:**

In this paper, the authors present a new approach for uncertainty estimation utilizing joint predictions. They introduce the concept of an Epistemic Neural Network (ENN). Within this framework, they propose an innovative architecture called 'epinet,' which supplements any conventional neural network. This additional architecture helps conventional neural networks outperform large ensembles in terms of the joint loglikelihood at the cost of a minimal computational increase. This approach is shown to be effective and practical through numerical experiments, including large-scale datasets like ImageNet.

**Strengths:**

1. The paper addresses an important problem of uncertainty quantification and disentanglement (aleatoric and epistemic) and introduces a novel class of models, ENN, to tackle it.
2. The paper is well-written and easy to follow.
3. The code for the reproduction of experiments is provided.

**Weaknesses:**

I think that there is a missing opportunity to show the performance of the proposed approach on direct uncertainty quantification. The motivation example in the introduction section showed that it is important to realise where uncertainty came from -- is it due to the noise in data (aleatoric) or due to the lack of knowledge (epistemic).
It would be helpful to see the authors run standard experiments, like checking how well the model can tell the difference between one dataset and another. This would give us a clearer picture of how well the model understands what it doesn't know. One such experiment might be to train a model on one dataset (say CIFAR100) and compare some epistemic uncertainty measure on another dataset (say LSUN). And then compute ROCAUC.
Right now, the results are just about how well the model can classify data and predict losses based on the same training dataset, which I believe doesn't tell us the whole story.

Minor concerns:
1) In the related work section, line 74, there are cited several papers, and mentioned their connection to Gaussian Processes. But two out of three cited papers - PriorNets and PostNets -- actually focus more on Dirichlet parametrization and don't have a clear link to Gaussian Processes.  I think SNGP[1] should be cited here.


[1] Liu, Jeremiah, et al. "Simple and principled uncertainty estimation with deterministic deep learning via distance awareness." Advances in Neural Information Processing Systems 33 (2020): 7498-7512.

**Questions:**

Please address the weaknesses I mentioned above.

Edit: I would like to thank the authors for their answers during rebuttal period.

**Limitations:**

Authors adequately addressed the limitations, there is no negative societal impact.

---

> ### Author Rebuttal · Authors · 2023-08-10
>
> We thank the reviewer for their time and effort in reviewing this paper. Our responses are:
>
> > I think that there is a missing opportunity to show the performance of the proposed approach on direct uncertainty quantification... It would be helpful to see the authors run standard experiments, like checking how well the model can tell the difference between one dataset and another...
>
> Thanks for the great suggestion!  We plan to include a visualization of the uncertainty estimates produced by our methods on some 2D OOD-like classification problems. This should give more intuition on how epinets are performing outside of the train dataset. It will be interesting to investigate more about how our methods will perform on more general OOD tasks  in the future.
>
> > In the related work section, line 74, there are cited several papers, and mentioned their connection to Gaussian Processes. But two out of three cited papers - PriorNets and PostNets -- actually focus more on Dirichlet parametrization and don't have a clear link to Gaussian Processes. I think SNGP[1] should be cited here.
>
> Thanks for pointing us to the SNGP paper [1]. We will make appropriate modifications and include the paper.

---

### Official Review · Reviewer_jzvR · 2023-07-07

**Soundness:** 3 good
**Presentation:** 4 excellent
**Contribution:** 3 good
**Rating:** 7
**Confidence:** 3

**Summary:**

The paper introduces epinets as part of Epistemic Neural Networks (ENNs), a novel approach to uncertainty estimation in deep learning models. Epinets extend neural networks to create ENNs, which can be used to estimate uncertainty. The paper then proceeds to present experiments on image classification tasks, demonstrating that epinets outperform both uncertainty baseline models and ensemble methods in terms of joint log-loss, while maintaining comparable performance in terms of marginal log-loss. At the same time, the computational cost of these networks is orders of magnitude less than larger ensembles.

**Strengths:**

- The paper presents a clear distinction between epistemic neural networks and Bayesian neural networks (BNNs). This distinction helps readers grasp the unique characteristics and advantages of the proposed framework.
- The results showcase the benefits of epinet compared to both ensemble approaches and uncertainty baselines. More comprehensive results in terms of the computational cost in FLOPs are provided in the appendix.
- The explanations provided throughout the paper are balanced between intuitive and easy-to-follow explanations and rigorous theorems. The figures provided in the document also effectively illustrate both the core principles of the epinet framework and the key results obtained. These visuals enhance the understanding of the concepts discussed in the paper.

**Weaknesses:**

- Even though the framework is rather general, the focus of the experiments seems rather narrow and confined to image classification tasks. It would be useful to see how well the epinet generalizes to other domains and whether there are scenarios or domains where it is more or less effective.
- Related to the previous point, the limitations of the work are not discussed in the conclusion. While epistemic neural networks are a general framework, addressing such limitations could provide insights on future directions to take.

**Questions:**

1. Has the epinet framework been tested on real-world datasets outside of the experiments mentioned in the paper?
2. How does the interpretability of the uncertainty estimates by epinets compare to that of BNNs?
3. In the ablation study in the appendix, it is shown that larger values for the index dimension improve both the joint and marginal kl estimates. Is there a point where this is no longer the case? What is the tradeoff?

**Limitations:**

To my knowledge, the limitations of epistemic neural networks are not addressed in the paper. The limitations and possible directions for future work could be addressed to improve the understanding and practicality of epinets.

Edit: I thank the authors for addressing my questions and comments in their rebuttal.

---

> ### Author Rebuttal · Authors · 2023-08-10
>
> We thank the reviewer for reviewing our paper. Our point-to-point responses are as follows:
>
> **Even though the framework is rather general, the focus of the experiments seems rather narrow and confined to image classification tasks. It would be useful to see how well the epinet generalizes to other domains and whether there are scenarios or domains where it is more or less effective.**
>
> We thank the reviewer for this comment. We would like to point the reviewer to a recent paper [Osband et al. 2023], on approximate Thompson sampling via epistemic neural networks. [Osband et al. 2023] has provided more experiment results for epinet, in both the neural bandit problems and reinforcement learning problems. Epinet has been found to be effective in these two problems.
>
> Osband et al. “Approximate Thompson sampling via Epistemic Neural Networks” https://arxiv.org/pdf/2302.09205.pdf
>
> **Related to the previous point, the limitations of the work are not discussed in the conclusion. While epistemic neural networks are a general framework, addressing such limitations could provide insights on future directions to take.**
>
> We thank the reviewer for this comment. One limitation of the current ENN framework is that it does not consider problems with sequential inputs and outputs (e.g. the language translation problems). We will clarify and discuss this limitation in the revision. A possible future direction is to build ENNs for these sequential problems.
>
> **Has the epinet framework been tested on real-world datasets outside of the experiments mentioned in the paper?**
>
> As we have mentioned above, the epinet framework has been tested in other problems, including the neural bandit problems and reinforcement learning problems.
>
> **How does the interpretability of the uncertainty estimates by epinets compare to that of BNNs?**
>
> Our understanding is that interpreting uncertainty modeling in complex prediction/decision problems might be too challenging. It is better to compare them based on the effectiveness of resulting predictions/decisions. In this paper, we have used the joint log-loss to measure the uncertainty estimates from different agents, including epinets and BNNs. As Theorem 2 of this paper shows, minimizing joint log-loss leads to effective decisions.
>
> **In the ablation study in the appendix, it is shown that larger values for the index dimension improve both the joint and marginal kl estimates. Is there a point where this is no longer the case? What is the tradeoff?**
>
> Thanks for the comment. Usually, increasing the ENN index dimension will improve both the joint and marginal kl estimates (performance). However, it will also require more computation. Thus, there exists a computation-performance tradeoff here.

---

### Official Review · Reviewer_TShA · 2023-07-10

**Soundness:** 3 good
**Presentation:** 2 fair
**Contribution:** 3 good
**Rating:** 5
**Confidence:** 2

**Summary:**

In this work, the authors introduce a new framework "Epistemic Networks" to better capture uncertainty by integrating over epistemic indices when computing the joint distribution over multiple inputs. This approach is flexible and can be added to any existing neural network architecture with some differences in the training algorithm. The authors show through experimental evaluations how their approach shows promising results when compared against other baseline methods for uncertainty quantification, including BNNs and deep ensembles on joint log-loss.

**Strengths:**

- This paper presents a new approach that's flexible and doesn't add a lot of parameters to the existing model.

- The paper evaluates the joint log-loss performance rather than just marginal log-loss and thus helps us understand how other baseline methods fare when looking at the joint prediction performance.

**Weaknesses:**

- I think the authors need to do a better job to motivate the case for joint log-loss. Why should one care for joint log-loss performance as opposed to marginal? Can you describe any applications where this might be useful?

- The presentation can improve a bit. The authors should describe section 4 to make sure it's easier to understand for folks. For e.g., what are all the different terms in sec 4.2 (eq 7, 8) and what do they signify?

**Questions:**

- Can you highlight the parallels between the expected reward under joint prediction to the work in the Bayesian decision theory area? In Bayesian decision theory too our goal is to get better decision utility for downstream decision-making task. [1-3]

- Building upon the previous point, why should one focus on the decision making framework presented in this paper versus other Bayesian decision theoretic applications that look at marginals? [1-3]

- Could you you describe how this approach captures aleatoric uncertainty more clearly?

References

[1] Vadera, Meet, Soumya Ghosh, Kenney Ng, and Benjamin M. Marlin. "Post-hoc loss-calibration for Bayesian neural networks." In UAI (2021).

[2] Cobb, Adam D., Stephen J. Roberts, and Yarin Gal. "Loss-calibrated approximate inference in Bayesian neural networks." arXiv preprint arXiv:1805.03901 (2018).

[3] Simon Lacoste-Julien, Ferenc Huszár, and Zoubin Ghahramani. Approximate inference for the loss-calibrated
bayesian. In AISTATS (2011)


**Limitations:**

Highlighted above in the review. The authors do not clearly specify the limitations in their submission.

---

> ### Author Rebuttal · Authors · 2023-08-10
>
> We would like to thank Reviewer TShA for reviewing our paper. The following are our point-to-point responses:
>
> **I think the authors need to do a better job to motivate the case for joint log-loss. Why should one care for joint log-loss performance as opposed to marginal? Can you describe any applications where this might be useful?**
>
> We would like to clarify that, as a recent paper [Wen et al. 2022] has shown, for a broad class of decision problems, such as combinatorial decision problems, sequential prediction problems, and multi-armed bandit problems, accurate joint predictions are required to deliver good performance, while accurate marginal predictions alone are insufficient to guarantee good performance. We will further explain and discuss this in the revision.
>
> Wen et al. 2022, “From predictions to decisions: the importance of joint predictive distributions ”, https://arxiv.org/abs/2107.09224.
>
> **The presentation can improve a bit. The authors should describe section 4 to make sure it's easier to understand for folks. For e.g., what are all the different terms in sec 4.2 (eq 7, 8) and what do they signify?**
>
> Thanks for raising this. We will further improve the writing of Section 4.2. The notations in equations 7 and 8 follow the definitions in Sections 3 and 4.1. In particular: \theta is the parameters of the ENN, z is the ENN index, (x_i, y_i) is an input-label pair. f_\theta(x, z_i) is the output of the ENN, which is the logit vector. The softmax transforms the logit vector into a probability vector, and the subscript y_i  in eq 7 denotes the (y_i)-th component of this probability vector. We will clarify that in eq 8, \Phi is the CDF of a standard Gaussian random variable.
>
> We recognize that Section 4.2 only explicitly states the loss for a single input-label pair and a single epistemic index. We will clarify in the paper that for each stochastic gradient step, the method samples a batch of input-label pairs and a batch of indices and averages over the losses.
>
> **Can you highlight the parallels between the expected reward under joint prediction to the work in the Bayesian decision theory area? In Bayesian decision theory too our goal is to get better decision utility for downstream decision-making task. [1-3]**
>
> Bayesian decision problems are typically framed as maximizing expected utility.  Our approach offers tools for such problems.  Theorem 2 ensures that minimizing joint log-loss leads to effective decisions for problems framed in that manner.
>
> Please also note that, as Theorem 2 indicates, for the special case when the reward only depends on one label (i.e. tau=1 in the formal version of Theorem 2 in Appendix B), joint loss reduces to marginal loss. For that special case, minimizing marginal log-loss is sufficient.
>
> **Building upon the previous point, why should one focus on the decision making framework presented in this paper versus other Bayesian decision theoretic applications that look at marginals? [1-3]**
>
> As we have discussed above, our approach is consistent with Bayesian decision theory.  Joint log-loss serves as a unit test to ensure that our tool will serve the needs of Bayesian decision making.
>
> **Could you you describe how this approach captures aleatoric uncertainty more clearly?**
>
> Variation across epistemic indices models epistemic uncertainty.  For a fixed epistemic index, label probabilities produced as model outputs express aleatoric uncertainty.

---

> > ### Comment · Reviewer_TShA · 2023-08-18
> > **Response to authors**
> >
> > I thank the authors for their rebuttal. I've looked at their rebuttal as well as other reviews on the paper.
> >
> > Furthermore, I've come to realize that my initial assessment might have fallen short - I am reducing my confidence on the review and increasing the score based on authors' response.
> >
> > Nonetheless, the paper does need significant updates to its presentation, and this has also been pointed out by one other reviewer.

---

### Official Review · Reviewer_UZs5 · 2023-07-27

**Soundness:** 3 good
**Presentation:** 3 good
**Contribution:** 3 good
**Rating:** 7
**Confidence:** 4

**Summary:**

This paper proposes epistemic neural networks (ENNs) as a novel framework for uncertainty estimation in neural network predictions. ENNs introduce an epistemic index that expresses uncertainty and correlations across multiple inputs via joint predictive distributions. The paper argues that joint predictions are critical for effectively evaluating uncertainty quality and enabling good decision making. It is argued that minimizing joint log-loss leads to near optimal actions while marginal log-loss does not. ENNs generalize Bayesian neural networks, as any BNN can be expressed as an ENN but the reverse does not hold. A novel ENN architecture called epinet is introduced, which adds a small auxiliary network to any existing neural network to produce uncertainty estimates. Experiments demonstrate that epinets can match the joint prediction performance of large ensembles while adding relatively minor computational overhead.

**Strengths:**

Originality:

Proposes a new conceptual framework of epistemic neural networks that expands uncertainty estimation options beyond Bayesian neural networks.

Introduces a practically effective and scalable approach via epinets that represents a novel architecture and training methodology.

Quality:

Technically sound way of modeling uncertainty through joint predictive distributions.

Strong empirical results outperforming baselines.

Clarity:

Motivates the limitations of marginal predictions and need for joint modeling.

Explains and visualizes key concepts effectively through examples.

Results presented clearly through tables/plots.

Significance:

Uncertainty estimation is a fundamental challenge for deploying reliable neural networks.

Can enable progress on critical applications like exploration, experiment design, and robust decision making.



**Weaknesses:**

Theoretical Analysis:

The connections between joint log-loss and decision performance could be bolstered with more rigorous analysis of regret bounds or performance guarantees. This would strengthen claims about utility for decision making.

Additional theoretical characterization of the proposed training objectives and epinet architecture properties could provide better insights into why the approach works.

Experimental Evaluation:

Experiments focus on image classification. Evaluating on decision-making benchmarks more directly relevant to the motivations could better highlight benefits.

Could ablate design decisions like network architectures and priors more thoroughly to understand their impact. This provides guidance on best practices.

Novelty and Impact:

The high-level ENN concept builds closely on established perspectives like Bayesian learning. Examining more novel implications would strengthen novelty.

While solid incremental gains are shown, the broader advancements enabled by this approach could be better highlighted. Articulating impact on future work would increase significance.

**Questions:**

Theoretical Analysis

Could you provide a more thorough theoretical analysis quantifying the advantages of joint modeling for decision making? Any regret bounds or performance guarantees?

Is it possible to better characterize the properties of the proposed training objectives? Do they provably optimize calibration of uncertainties?

Experimental Evaluation

Have you considered evaluating on tasks more directly relevant to decision making such as experiment design, active learning, or contextual bandits?


Impact

Could you better highlight the broader potential impact enabled by the approach? What new directions are unlocked?

---

> ### Author Rebuttal · Authors · 2023-08-10
>
> We would like to thank the reviewer for their time and effort in reviewing the paper. We address the comments below:
>
> **> Theoretical analysis regarding regret bound: Could you provide a more thorough theoretical analysis quantifying the advantages of joint modeling for decision making? Any regret bounds or performance guarantees?**
>
> Thanks for raising this. We would like to point the author to Theorem 2, which has a formal version and a proof in Appendix B. The theorem establishes a rigorous regret bound in bandit settings. It is an adaptation of the theory presented by Wen et al. (https://arxiv.org/pdf/2107.09224.pdf), which we plan to cite in the main paper as well.
>
> **> Theory regarding the choice of training objective: Is it possible to better characterize the properties of the proposed training objectives? Do they provably optimize calibration of uncertainties?**
>
> We would like to point the reviewer to Theorem 4 and the associated lemmas and proof in Appendix D. In Theorem 4, we show that the distribution approximated by an epinet converges to the posterior in linear regression setting, under appropriate technical conditions. We plan to extend this theory beyond linear regression in future work.
>
> **> Have you considered evaluating on tasks more directly relevant to decision making such as experiment design, active learning, or contextual bandits?**
>
> We would like to point the reviewer to a relevant paper: (https://arxiv.org/pdf/2302.09205.pdf)  which presents empirical results that improved joint predictions via epinet lead to better performance in bandit and reinforcement learning tasks.
>
> **> Could you better highlight the broader potential impact enabled by the approach? What new directions are unlocked?**
>
> Our main motivation for this work is to enable principled and practical methods for scalable uncertainty estimation. Our work, especially epinet, will enable uncertainty estimation for large scale models (such as large language models). Furthermore, scalable uncertainty estimation can also unlock practical implementations of efficient exploration algorithms, such as Thompson sampling and information-directed sampling, which may dramatically improve data efficiency in sequential decision-making tasks. Finally, our ENN framework and open source library will allow researchers to iterate and develop better networks and approaches to uncertainty modeling.

---

### Decision · Program_Chairs · 2023-09-21

**Decision:**

Accept (spotlight)

**Comment:**

The reviewers praise the significance, originality, strong empirical results, technical soundness, and flexibility and scalability of the approach. The main criticisms are the lack of theoretical insights, limited scope of experiments, lack of motivation, lack of clarity in the presentation, and lack of OOD experiments. However, these concerns have been mostly addressed by the rebuttal.